# Dark Diazotrophy during the Late Summer in Surface Waters of Chile Bay, West Antarctic Peninsula

**DOI:** 10.3390/microorganisms10061140

**Published:** 2022-05-31

**Authors:** María E. Alcamán-Arias, Jerónimo Cifuentes-Anticevic, Wilson Castillo-Inaipil, Laura Farías, Cynthia Sanhueza, Beatriz Fernández-Gómez, Josefa Verdugo, Leslie Abarzua, Christina Ridley, Javier Tamayo-Leiva, Beatriz Díez

**Affiliations:** 1Departamento de Oceanografía, Universidad de Concepción, Concepción 4030000, Chile; mealcaman@uc.cl (M.E.A.-A.); laura.farias@udec.cl (L.F.); leslie.abarzua.ortiz@gmail.com (L.A.); 2Center for Climate and Resilience Research (CR)2, Universidad de Chile, Blanco Encalada 2002, Santiago 8320000, Chile; christina.m.ridley@gmail.com (C.R.); javierignacio.tamayo@gmail.com (J.T.-L.); 3Escuela de Medicina, Universidad Espíritu Santo, Guayaquil 0901952, Ecuador; 4Departamento de Genética Molecular y Microbiología, Pontificia Universidad Católica de Chile, Santiago 8331150, Chile; jeronimo.cifuentes@ug.uchile.cl (J.C.-A.); wilson.castillo@ug.uchile.cl (W.C.-I.); ctsanhueza@gmail.com (C.S.); beatriz.fernandez@ulpgc.es (B.F.-G.); 5Instituto de Oceanografía y Cambio Global (IOCAG), Universidad de Las Palmas de Gran Canaria (ULPGC), 35001 Las Palmas, Spain; 6Alfred-Wegener-Institute, Helmholtz Centre for Polar and Marine Research, 27570 Bremerhaven, Germany; maria.josefa.verdugo@awi.de; 7Center for Genome Regulation (CRG), Universidad de Chile, Blanco Encalada 2085, Santiago 8320000, Chile

**Keywords:** nitrogen fixation, heterotrophic diazotrophy, WAP/new production, diazotrophy

## Abstract

Although crucial for the addition of new nitrogen in marine ecosystems, dinitrogen (N_2_) fixation remains an understudied process, especially under dark conditions and in polar coastal areas, such as the West Antarctic Peninsula (WAP). New measurements of light and dark N_2_ fixation rates in parallel with carbon (C) fixation rates, as well as analysis of the genetic marker *nif*H for diazotrophic organisms, were conducted during the late summer in the coastal waters of Chile Bay, South Shetland Islands, WAP. During six late summers (February 2013 to 2019), Chile Bay was characterized by high NO_3_^−^ concentrations (~20 µM) and an NH_4_^+^ content that remained stable near 0.5 µM. The N:P ratio was approximately 14.1, thus close to that of the Redfield ratio (16:1). The presence of Cluster I and Cluster III *nif*H gene sequences closely related to Alpha-, Delta- and, to a lesser extent, Gammaproteobacteria, suggests that chemosynthetic and heterotrophic bacteria are primarily responsible for N_2_ fixation in the bay. Photosynthetic carbon assimilation ranged from 51.18 to 1471 nmol C L^−1^ d^−1^, while dark chemosynthesis ranged from 9.24 to 805 nmol C L^−1^ d^−1^. N_2_ fixation rates were higher under dark conditions (up to 45.40 nmol N L^−1^ d^−1^) than under light conditions (up to 7.70 nmol N L^−1^ d^−1^), possibly contributing more than 37% to new nitrogen-based production (≥2.5 g N m^−2^ y^−1^). Of all the environmental factors measured, only PO_4_^3-^ exhibited a significant correlation with C and N_2_ rates, being negatively correlated (*p* < 0.05) with dark chemosynthesis and N_2_ fixation under the light condition, revealing the importance of the N:P ratio for these processes in Chile Bay. This significant contribution of N_2_ fixation expands the ubiquity and biological potential of these marine chemosynthetic diazotrophs. As such, this process should be considered along with the entire N cycle when further reviewing highly productive Antarctic coastal waters and the diazotrophic potential of the global marine ecosystem.

## 1. Introduction

Nitrogen is a limiting nutrient in most open ocean regions. Marine dinitrogen (N_2_) fixation is an essential process that biologically transforms atmospheric N_2_ into bioavailable ammonium (NH_4_^+^), which is important for evaluating the global role of the upper ocean in atmospheric CO_2_ sequestration [1,2]. To date, most studies on N_2_ fixation rates and diazotrophic distribution have been conducted in tropical and subtropical open oceans [3,4,5,6,7,8,9]. However, in low-temperature waters, such as those of the Arctic Ocean, predictions estimate that N_2_ fixation rates could contribute up to 3.5 Tg N y^−1^ to the nitrogen budget [10]. In an ice-free Arctic scenario, this would represent approximately 9.2 Tg N y^−1^, thus contributing 7.1% of the fixed N_2_ in the global marine system [10]; however, these estimates are variable across the Arctic Ocean [11,12,13]. In the Southern Ocean, the lack of estimated N_2_ fixation rates directly interferes with the ability to calculate regional and global marine nitrogen budgets and hinders the identification of the main factors controlling marine N_2_ fixation [11,14].

To date, different factors, such as oxygen, light, temperature, inorganic nitrogen, phosphorus, iron, organic matter and trace metal availability, have been reported as variables that potentially control both N_2_ fixation and the distribution of diazotrophs in marine systems [5,8,15,16]. Marine diazotrophs, traditionally assessed based on the detection of the *nif*H gene, are shown to be primarily associated with Cyanobacteria [17,18], but there is also substantial evidence for a high diversity and wide distribution of non-cyanobacterial diazotrophs (Bacteria and Archaea) in the open ocean [19,20,21,22,23] and in coastal and estuarine environments [4]. The discovery of new marine niches for N_2_ fixation has been accompanied by a growing appreciation for the enormous diversity of N_2_-fixing species, including those involved in diatom–diazotroph associations and heterotrophic diazotrophs that are primarily widespread in the sunlit marine layer [9,22,23,24,25,26,27,28]. In less explored marine environments, such as the polar regions, the abundance of diazotrophs has been reported in Arctic sea ice and seawater [12,13,29,30,31], while symbiosis between N_2_-fixing cyanobacteria (UCYN-A) and haptophytes has been described in the Western Arctic and Bering Sea [10]. In addition, non-cyanobacterial diazotrophs belonging to heterotrophic Alphaproteobacteria, Deltaproteobacteria, Epsilonproteobacteria and Actinobacteria, along with some unclassified bacteria, have been observed in Arctic sea ice and seawater [12,13,30], as well as at some stations south of 59° S [32] and in the eastern part of the Southern Ocean [11]. Nevertheless, for sub-Antarctic areas of the Southern Ocean, both N_2_ fixation rates and taxonomic characterization of the *nif*H gene have been reported along the 170° W meridian between 66° S and 52° S (Southern Pacific Ocean) [32] and at 50° S in the Southern Indian Ocean [33]. Models predict an absence of biological nitrogen fixation in Antarctic waters due to limited iron availability and high nitrogen concentrations [34]. However, high nitrogen fixation rates (44 nmol N L^−1^ d^−1^) and the presence of diazotrophs, such as UCYN-A (*Candidatus* Atelocyanobacterium thalassa), have been detected near the Antarctic coast, especially around ice-covered regions [11]. Despite this, neither nitrogen fixation rates nor the occurrence of diazotrophs has yet been reported for the West Antarctic Peninsula (WAP), limiting our knowledge of the actual geographic distribution of diazotrophic activity in the Antarctic Ocean.

Here, we report the first N_2_ fixation rates over six consecutive late summers (2013, 2014, 2016–2019) in the surface layer (up to 30 m depth) of the coastal waters of Chile Bay, located in the South Shetland Islands, WAP. We also estimated the rates of contribution to new nitrogen-based production, and for late summer 2019 we revealed the taxonomic identity of potential nitrogen fixers in these Western Antarctic waters both at 2 and 30 m. Overall, the results provide novel and invaluable information that contributes to global records on diazotrophy, thereby allowing a better understanding of nitrogen balances and diazotrophic potential in the global marine ecosystem.

## 2. Materials and Methods

### 2.1. Sampling Site and Environmental Variable Measurements

A specific marine sampling point (62°27′6″ S, 59°40′6″ W; station P3, Figure 1) in Chile Bay (Greenwich Island, South Shetland Islands, WAP) was monitored over six late summer periods (8 February 2013; 14 and 22 February 2014; 22 February 2016; 10 and 21 February 2017; 9 and 16 February 2018; and 17 February and 8 March 2019). Seawater from 2 m (all years) and 30 m depths (2017–2019) were sampled using a surface hand pump and Niskin bottle, respectively, to determine N_2_ fixation rates as part of a marine biogeochemical monitoring time series. Seawater temperature (°C) and salinity (PSU) were measured in 2013, 2014 and 2016 with a multi-parameter sensor (OAKTON PCD650), while in 2017, 2018 and 2019, physical variables were measured with a Conductivity Temperature Depth profiler (CTD; SeaBird19 plus). For chlorophyll *a* (Chl-*a*) measurements, triplicate 1-L samples of seawater from both 2 and 30 m depth were prefiltered with a 150-µm net to exclude large organisms, and then the remaining biomass was collected by subsequent filtration with 0.7-µm GF/F glass fiber filters. Each filter was frozen until processing, using methanol (2013–2014) or acetone (2016–2019) extraction protocols, after which extracts were analyzed by spectrophotometric [35] or fluorometric [36] techniques, respectively. Samples for the determination of inorganic nutrients (nitrite (NO_2_^−^), nitrate (NO_3_^−^) and phosphate (PO_4_^3−^)) were also taken in triplicate at 2 and 30 m (only for 2017–2019), stored in 15 mL polyethylene tubes and frozen at −20 °C in the dark until analysis. NO_2_^−^, NO_3_^−^ and PO_4_^−3^ concentrations were measured with a Seal AutoAnalyzer 3 (AA3, SEAL Analytical, Mequon, WI, USA) [37]. Ammonium (NH_4_^+^) concentrations were only measured for 2017, 2018 and 2019, using the method of [38] with a minor modification. Briefly, 20 mL of seawater (taken directly from the Niskin bottle) was aliquoted in triplicate into 50-mL borosilicate glass vials and fixed with the working solution. After 4 h in the dark at ambient temperature (~1–2 °C), the flasks were incubated for 2 h at 40 °C. The absorbance was then measured fluorometrically (AquaFluor Handheld, Turner Designs). The inorganic N:P ratio was calculated using the resulting nitrogen species (NO_3_^−^ + NO_2_^−^ + NH_4_^+^) and phosphate concentrations.

### 2.2. Phylogenetic Characterization of the Diazotrophic Community

Phylogenetic characterization of diazotrophs potentially responsible for N_2_ fixation in Chile Bay during late summer 2019 was performed. For this purpose, seawater (3-L) was collected from station P3 at 2 and 30 m depth in parallel for the N_2_ and C uptake experiments. Seawater was pre-filtered with 150-µm mesh and then filtered through 0.22-μm pore size filters (Sterivex units, Merck-Millipore, Burlington, MA, USA) using a peristaltic pump (Cole Palmer System Model no. 7553-70 (6–600 rpm)). Filters were immediately frozen at −20 °C and then stored at −80 °C in the laboratory until DNA extraction.

DNA was extracted according to the protocol described by [39] with some modifications. Briefly, after resuspending the filters in xanthogenate lysis buffer, DNA was extracted with phenol–chloroform–isoamyl alcohol (25:24:1) and then the residual phenol was removed with chloroform–isoamyl alcohol (24:1). The extracts were cleaned by overnight precipitation with cold isopropanol and washed with 70% ethanol. DNA was quantified using a Qubit^®^ 2.0 Fluorometer (Thermo Fisher Scientific, Waltham, MA, USA). Quality was assessed by spectrophotometry (A260/A280 ratio) and integrity was checked by standard agarose gel electrophoresis.

The identity of N_2_-fixing Bacteria and Archaea in the coastal waters of Chile Bay was determined by high-throughput *nif*H amplicon sequencing. A composite sample of marine DNA from 2 and 30 m depth was used as the template to obtain *nif*H amplicons using two different primer pairs: PolF-PolR [40] and Ueda19F-R6 [41]. Sequencing was performed on the Illumina MiSeq platform (2 × 250 bp) at MR DNA Laboratories (Shallowater, TX, USA). For each primer set, raw sequences of the *nif*H gene were trimmed with PRINSEQ [42]. Paired-end sequences were imported into QIIME2 and joined using DADA2 to obtain amplicon sequence variants (ASVs) [43,44]. ASVs were translated into amino acid sequences using Prodigal [45], clustered to 97% identity using cd-hit [46] and curated based on the Hidden Markov Model profiles described in NifMAP [41] to filter out homologous *nif*H genes (*bch*L, *chl*L, *bch*X, *par*A).

For phylogenetic placement of NifH amino acid sequences, we build a NifH amino acid database from the NCBI non-redundant (NR) database. NifH reference sequences were aligned using MAFFT [47], the multiple sequence alignment was manually refined and the phylogenetic tree was calculated with iQTree (-bb 10,000-alrt 10,000-nm 10,000-m WAG) [48,49]. Chile Bay NifH amino acid sequences were clustered to 97% identity from ASVs with a relative abundance >1% (NifH OTUs), and short NifH sequences from the best-hit BLASTN (June 2020 NCBI NT database) were aligned to the reference NifH multiple sequence alignment using MAFFT (--keeplength --add) [47] and then phylogenetically placed into the reference NifH tree using the EPA-ng algorithm [50]. The resulting tree was processed with GAPPA [51] and visualized with iTOL v6.3 [52]. All *nif*H sequences were deposited in the NCBI database under accession numbers MZ485323-MZ485340.

To assess the presence of potentially chemoautotrophic diazotrophs in Chile Bay coastal waters, we performed a literature search for the reference microorganisms used to build the phylogenetic reference tree. Next, each organism was classified according to its metabolic class, and finally, the identified chemoautotrophs closely related to the Chile Bay *nif*H sequences were recorded in the Appendix A. In addition, a BLASTP search was made to obtain the percentage of identical matches and e-value between the chemoautotrophs and the Chile Bay OTUs of NifH proteins (Appendix A).

### 2.3. Nitrogen Fixation Assays

In the late summer of 2013, 2014 and 2016–2019, isotopic nitrogen (^15^N_2_) and carbon (H^13^CO_3_^−^) uptake were measured in the same incubations to jointly unveil the contribution of the N_2_ fixation rates of local primary production (N_2_LPP) and nitrogen-based new production (N_2_NP). These nitrogen and carbon uptake assays were conducted in polycarbonate bottles (2.7-L) using 2 m deep seawater for all years and 30 m deep seawater for the years 2017–2019. For samples taken at 2 m, uncovered transparent polycarbonate bottles were incubated in triplicate, while another triplicate set of covered bottles (using a black bag and aluminum foil) was used to obtain information on N_2_ fixation in the absence of light by the non-photosynthetic community. For all 30 m samples, covered bottles (using a black bag and aluminum foil) were incubated in triplicate to simulate the dark conditions that exist during the day time at this depth (1% of the PAR only reaches <20 m in the bay [53]). Due to the complicated logistics, it was not possible to perform these incubations directly offshore at station P3; thus, all incubations were performed at 2 m depth in near-shore bay water (average seawater temperature 1–1.5 °C) for a period of 24 h to obtain a daily rate. ^15^N uptake was initiated by adding 2 mL of ^15^N_2_ gas (98% atom ^15^N_2_ gas; Sigma-Aldrich, St. Louis, MO, USA (SZ1670) for 2013–2014 years and Cambridge Isotope Laboratories Inc. for 2016–2019 years) into each bottle sealed with a silicone septum, using a 5 mL gas-tight syringe. It is important to note that, years after performing these experiments, it was reported in the literature that part of the Sigma-Aldrich ^15^N_2_ gas batch (SZ1670) may contain high concentrations of inorganic nitrogen contamination [54,55,56]. Because of this, the rates obtained in this study for 2013 and 2014 in Chile Bay could reflect this measurement error. Unfortunately, due to a lack of remaining gas, it was not possible to quantify the errors by gas dilution in water or evaluate possible contamination of the standards of reference used for isotopic ratio mass spectrometry (IRMS), among other corrections reported by [55]. Nevertheless, the rates recorded from 2016–2019 were obtained using a stock of ^15^N_2_ gas from another company (Cambridge Isotope Laboratories) for which no contamination has been reported to date; therefore, the rates obtained for these years should not contain any error. To estimate carbon (H^13^CO_3_^−^) uptake by photoautotrophs (light) and chemoautotrophs (darkness), the same polycarbonate bottles injected with ^15^N_2_ gas at both depths and light conditions were amended with 2.5-mL of H^13^CO_3_^−^ (3.645 mg ^13^C mL^−1^). This concentration is equivalent to 0.5 μmol mL^−1^ and corresponds to an enrichment of ~10%, in agreement with natural dissolved inorganic carbon concentrations reported for the coastal WAP [57]. In addition, to calculate the contribution of N_2_ fixation to new nitrogen production (N_2_NP), nitrate assimilation rates were measured for each year. These assays were performed by adding 300 µL of K^15^NO_3_ (99% at 0.5 μmol mL^−1^) in uncovered and covered 600 mL polycarbonate bottles containing seawater from 2 m (all years) and 30 m (2017–2019). A seawater control, with no isotope addition, was also incubated along with the bottle set to determine the natural isotopic composition (^15^N and ^13^C). After incubation (24 h), 1 L of water from the ^15^N_2_/H^13^CO_3_^−^ treatments and 600 mL from the K^15^NO_3_ treatments and controls were filtered through pre-combusted 0.7-µm GF/F glass fiber filters using a peristaltic pump. The filters were then frozen and kept at −20 °C until processing at the Universidad de Concepción, Chile. The amount of ^15^N_2_, H^13^CO_3_^−^ and K^15^NO_3_ assimilated during incubation, as well as the C:N ratio (composition of organic matter in the sample), were estimated by continuous-flow IRMS (Finnigan DELTAplus IRMS, Thermo Scientific), with a detection limit of 0.005 mg N and 0.074 mg C. Daily assimilation rates of ^15^N (ρN_2_fix) were calculated as described by [58], and the rates of ^13^C (ρ^13^C) and ^15^NO_3_ (ρ^15^NO_3_) assimilation were determined according to [39], using the particulate organic carbon and nitrogen (POC and PON, respectively) and % atm^15^N and ^13^C after the incubations.

### 2.4. N_2_ Fixation Contribution to Coastal Marine Productivity

The contribution of marine N_2_ fixation rates (ρN_2_fix) to new production (N_2_NP) was calculated according to [59], which included nitrate assimilation rates (ρ^15^NO_3_) (obtained at the same time during the dual ^15^N_2_/H^13^CO_3_- incubations), but excluded the nitrification rate correction (data not available). The contribution of fixed nitrogen to local primary production (N_2_LPP) was also estimated. For each year sampled, nitrogen (ρN_2_fix) and carbon (ρ^13^C) assimilation rates were calculated along with the molC:molN (C:N) ratio obtained from the IRMS analysis. Estimates of N_2_LPP and N_2_NP contributions were calculated using Equations (1) and (2), respectively:N_2_LPP = [(C:N × ρN_2_fix) × 100]/ρ^13^C(1)
N_2_NP = [(ρN_2_fix × 100)/(ρ^15^NO_3_ + ρN_2_fix)](2)

### 2.5. Statistical Analysis

To determine potential drivers modulating late summer N_2_ fixation and carbon assimilation activities in Chile Bay, we performed linear regressions between the obtained N_2_ and C fixation rates and the biogeochemical variables of temperature, Chl-*a*, NO_3_^−^, NO_2_^−^, NH_4_^+^ and PO_4_^3−^. The statistical analyses were carried out in R using the packages Stats (R Core Team, 2020, Vienna, Austria) and Vegan [60]. Estimates of linear relationships between continuous factors were analyzed through linear models, with axes transformed to logarithmic scale and clustered by different experimental conditions (i.e., light, darkness) (df = 1).

The significant differences between light and dark nitrogen and carbon fixation rates were resolved by t-tests at a significance level of 0.05% using Xlstat statistical software.

## 3. Results

### 3.1. Physical and Biogeochemical Variables of Late Summer Seawater in Chile Bay

The environmental variables recorded in Chile Bay during February and early March 2013, 2014 and 2016–2019 at station P3 (Figure 1), when samples were collected to perform N_2_ fixation measurements, are shown in Table 1. Seawater temperatures above 0 °C were recorded during most sampling events for all summers, reaching maximum values of 2.23 °C at 2 m and 1.42 °C at 30 m for the same year (February 2017). The coldest temperatures (−0.25 to 0.34 °C) were recorded during 2014 (February 2014) and 2016 sampling, probably due to the use of a less sensitive sensor compared to the other sampled years. On average the seawater temperatures were 0.8 °C and 1.2 °C at 2 and 30 m, respectively. Average salinity values reached 33.9 in the surface samples and 34.14 at 30 m, denoting a shallow haline stratification as reported recently [53]. Variability in Chl-*a* concentrations was detected both during and between summers for the monitored days. The lowest Chl-*a* concentration (0.03 mg L^−1^) was recorded in the 2013 sample at 2 m, increasing to 3.58 and 4.17 mg L^−1^ in 2018 and 2019, respectively, at the same depth (2 m). These differences are probably due to the different protocols used, with the fluorometric method [36] used in years 2016–2019 being much more sensitive [61].

The lowest detected NH_4_^+^ concentration was 0.33 µM in the 2017 sample. Furthermore, the NH_4_^+^ concentration in the surface layer was lower than that at 30 m depth for each sampling date, except those in 2018 and 2019, which both showed similar concentrations at both depths (Table 1). The NO_2_^−^ concentration reached the minimum value (0.13 µM) in the 2014 samples and the maximum value (0.32 µM) in the 2013 samples, showing little variation between depths. Meanwhile, NO_3_^−^ showed slight variation between monitored depths and years, recording the lowest value (14.56 µM) in the samples from 2016 at 2 m and the highest (23.30 µM) in 2017 (21 February) at 30 m. PO_4_^3−^ experienced little change between years, with the lowest values being 1.04 µM (16 February 2018; 2 m) and 1.13 µM (8 March 2019; 30 m). For the remaining years, the PO_4_^3−^ concentrations ranged from 1.19 µM (17 February 2019; 2 m) to 1.62 µM (8 February 2013; 2 m), with a concentration of 1.61 µM recorded at both 2 m (14 February 2014) and 30 m (21 February 2017) (Table 1). The N:P ratio ranged from 9.45 (2 m, 2016) to 18.22 (2 m, 2018), although most of the time it remained close to the Redfield ratio of 16 [62], as seen for 2019 at 30 m (N:P 16.52) (Table 1).

### 3.2. Phylogenetic Characterization of the Diazotrophic Community of Chile Bay

Phylogenetic reconstruction of *nif*H gene sequences, which were amplified using two different primer sets and template DNA from a mixture of 2 and 30 m samples from 2019 (see methodology), allowed us to identify the presence of diazotrophic organisms potentially responsible for the N_2_ fixation observed in Chile Bay during that late summer period. From the raw sequencing data, we proceeded to trim the reads for quality and then the amino acid sequences were curated against HMM models described in NifMAP to remove *nif*H homologs (*bch*L, *chl*L, *bch*X, *par*A). For the PolF-PolR primer set, this process resulted in a total of 2077 final reads that could be assigned from the initial 4948 reads, with 58% of the reads removed after filtration). In the case of the Ueda19F-R6 primer set, 1377 reads were obtained that could be assigned from the initial 2830 reads, with 64% of the reads removed after filtration). Those reads represent only 18 different *nif*H OTUs detected in Chile Bay and were positioned in a reference tree along with their closest best-hit sequences in BLAST, taxonomy and environmental source (Figure 2). In addition, these best-hit sequences (from 106 genomes) were analyzed bibliographically to recover their identity and energy source. From this literature survey, we ruled out the presence of phototrophic organisms, as well as UCYN-A bacteria, but identified the presence of some chemoautotrophs (Appendix A) that potentially represented up to 5.6% of the total recovered *nif*H sequences associated with diazotrophic organisms.

As shown in the dendrogram (Figure 2), Chile Bay *nif*H OTUs were affiliated with nitrogenase clusters I (Figure 2; red branch line) and III (Figure 2; dark blue branch line). Three of the OTUs (OTU1, OTU8 and OTU4) recovered using the PolF-R primer set accounted for 84.6% of the total reads retrieved with this primer set. These sequences belong to Cluster I and were closely related to heterotrophic and chemoautotrophic organisms, such as *Bradyrhizobium* of the class Alphaproteobacteria, and methanotrophs, such as *Methylocapsa, Methyloceanibacter, Methylocystis* and *Xanthobacter* (Figure 2; purple, Appendix A), which have been previously widely detected elsewhere in soil (outer ring, brown) and marine (blue) samples. OTUs retrieved with the second primer set Ueda19F–R6 were distributed along the entire *nif*H tree, being also mainly located in Cluster I, and closely associated with Deltaproteobacteria (order Desulfuromonadales; orange), Gammaproteobacteria (dark red) and Alphaproteobacteria (purple) from marine sediments (yellow) and soils (brown). As for nitrogenase Cluster III (dark blue), two OTUs were recovered using the PolF-R primer set and were closely related to Deltaproteobacteria (*Deltaproteobacteria bacterium*; OTU12) and Bacteroidetes (*Paludibacter*; OTU14).

### 3.3. N_2_ Fixation and Inorganic Carbon Assimilation Rates

The N_2_ fixation rates of the 2 m samples incubated under the light (uncovered) condition ranged from 0.35 to 7.70 nmol N L^−1^ d^−1^ (2017 and 2018, respectively), while those under the dark (covered) condition ranged from 0.48 to 24.51 nmol N L^−1^ d^−1^ (2016 and 2019, respectively) (Figure 3a; Table 2). Due to the possible contamination of the ^15^N_2_ gas batch used in the years 2013 and 2014, which could have affected their rates, we only mention here that they ranged from 0.45 to 1.03 nmol N L^−1^ d^−1^ (both at 2014) in uncovered bottles at 2 m and from 0.17 to 0.81 nmol N L^−1^ d^−1^ in covered bottles at 2 m. These rates are similar to those found in other years, such as 2016–2017. For the 2017–2019 incubations, dark rates retrieved in covered samples at 2 m were always significant and higher than the light rates (Figure 3a). For the 30 m samples (2017–2019, all incubated at 2 m close to the shore), the dark rates strongly fluctuated between years. The 2018 incubation showed the highest N_2_ fixation rates (45.40 nmol N L^−1^ d^−1^) recorded for this depth, which was significantly higher than the rates obtained for 2017 (*p* < 0.002; 2.39 nmol N L^−1^ d^−1^) and 2019 (*p* < 0.01; 1.21 nmol N L^−1^ d^−1^).

Phototrophic activity, measured in tandem with ^15^N_2_ fixation, showed the highest rates fluctuating between 837 and 1380 nmol C L^−1^ d^−1^ (Figure 3a, Table 2) for the 2 m samples (2016 to 2019) under the light condition (uncovered samples). Significant differences for the years 2013 (*t*-test *p* < 0.003) and 2014 (*t*-test *p* < 0.01) (Figure 3b) were detected between the photosynthetic (uncovered) and chemoautotrophic (covered) 2 m samples. Chemoautotrophic carbon fixation (covered samples) for both 2 and 30 m samples was always one order of magnitude lower than under the photosynthetic condition (uncovered samples), in general reaching >30% contribution to the total carbon fixation. Furthermore, dark rates at 30 m showed significant differences between 2017–2018 (*t*-test *p* < 0.02), 2017–2019 (*t*-test *p* < 0.008) and 2018–2019 (*t*-test *p* < 0.01).

Correlation analysis between N_2_ and C fixation rates with the measured environmental variables (temperature, Chl-*a*, NO_3_^−^, NO_2_^−^, NH_4_^+^ and PO_4_^3−^) was conducted by linear regression. Among all the evaluated variables, the results only revealed a significant limiting effect for the PO_4_^3−^ concentration on both rates under contrasting light and dark incubations. In particular, the data showed a marked negative correlation between N_2_ fixation and the PO_4_^3−^ concentration, observed exclusively under the light condition (R^2^ = 0.73; *p* = 0.007) (Figure 4a). In contrast, carbon assimilation rates showed a moderate negative correlation with the PO_4_^3−^ concentration exclusively under the dark condition (R^2^ = 0.65; *p =* 0.003) (Figure 4b).

### 3.4. Contribution of N_2_ Fixation to New Nitrogen in Chile Bay

Since N_2_ fixation is a process that acts as a source of new nitrogen in the marine system, we calculated the total dinitrogen-based new production (N_2_NP) in Chile Bay. As *nif*H OTUs revealed a low (<5.6%) presence of chemoautotrophic bacteria with N_2_ fixation capacity, the contribution of fixed N_2_ to total local primary production (N_2_LPP) was not determined. Mean molC:molN ratios of 6.10 and 4.83 were obtained for the light and dark incubations, respectively, with no differences between 2 and 30 m depths (Table 2). The contribution of N_2_ fixation to N_2_NP ranged from 0.31% (2017) to 9.82% (2016) at 2 m in uncovered samples, while maximum values for covered samples were 37.59% at 2 m (2016) and 12.88% at 30 m (2018).

## 4. Discussion

The global oceanic nitrogen budget remains controversial and unbalanced [5,63] and is far from being resolved [18,64]. This is mainly the result of a substantial underestimation of N_2_ fixation, due to different applied methodologies [65,66] and the limited sampling of many oceanic and coastal locations, such as those in polar oceans, where low or null biological N_2_ fixation has been predicted given the low iron and high nitrogen concentrations [34].

We know that high rates of nitrogen fixation and diazotrophs can occur in Antarctic coastal waters [11] and that some N_2_ fixation rates have been measured in illuminated open ocean areas of the Southern Ocean [32,33]. However, the western region of the Antarctic Peninsula, which is of particular importance for C sequestration [67,68,69], has so far been largely ignored with respect to the estimation of N_2_ fixation rates, as have non-illuminated waters, where the characterization of the diazotrophic community associated with N_2_ and C fixation rates have never been performed. In addition, recent efforts have been made to reach a consensus on a method to measure diazotrophic production in pelagic ecosystems [55], and new insights are being gained into the distribution of active diazotrophic cyanobacteria and other non-cyanobacterial diazotrophs in low-temperature waters, and coastal and upwelling oceanic areas [2,26].

In this study, we focused on obtaining N_2_ fixation rates and determining their potential contribution to new nitrogen production in Chile Bay, as well as identifying diazotrophs that may potentially contribute to this relevant de novo nitrogen production over consecutive summers along the WAP coast. Accordingly, we studied N_2_ fixation over six late summer periods in the coastal waters of Chile Bay, a characteristically nutrient-rich Antarctic marine system with variable Chl-*a* concentrations throughout and over different summers caused by bloom events [69,70].

### 4.1. Diazotrophs in Chile Bay

Research on the genetic taxonomy and distribution of diazotrophic organisms in the ocean remains a challenge. In particular, investigating the identity and distribution of diazotrophs in eastern Antarctic coastal seawater has only recently started [11]. In that sense, our study in Chile Bay (Western Antarctic coastal seawater) contributes new knowledge to diazotrophy in the WAP. In this bay, we previously characterized the water column and the identity and dynamics of phytoplankton, bacterioplankton and virioplankton under different productivity conditions during the summertime [39,53,70,71]. Our observations show that bacterioplankton were less diverse and dominant during low productivity periods compared to the highly diverse bacterioplankton community associated with a eukaryotic photoautotrophic bloom [70]. In addition, previous work has described the activity of the microbial community and its contribution to relevant ecological functions, mainly associated with biogeochemical C cycling (diatoms) and N assimilation (bacterioplankton) during bloom events [39,53], and the effect of environmental variables, such as changes in salinity due to glacial melt, and the shallow haline stratification on these (active) microbial communities has been investigated [53,69].

Through analysis of the *nif*H gene, this study in Chile Bay has enabled us to reveal the presence and identity of chemosynthetic and heterotrophic bacteria, as well as some chemoautotrophic methanogens, in the cold waters of Western Antarctica, as confirmed by sequences closely related to the cosmopolitan diazotroph groups of Alphaproteobacteria (Cluster I) and Deltaproteobacteria (Cluster III). These diazotrophs have recently been found worldwide in hundreds of metagenomes from the Tara Ocean project [28], and in the Eastern Antarctic Ocean [11] and the Central Arctic Ocean [13], confirming their presence in polar regions. In addition, diverse and active heterotrophic and chemoautotrophic diazotrophs have been found in sinking particles [72], planktonic aggregates [73], aphotic waters [22,74,75], oxygen minimum zones [76,77,78,79] and NH_4_^+^-rich sulfidic-anoxic waters of the Baltic Sea [80], where these *nif*H sequences were mainly related to anaerobic bacteria [11,22]. The *nif*H sequences related to the Cluster I nitrogenase found in Chile Bay also indicate the potential presence in these Antarctic waters of anaerobic heterotrophic diazotrophs associated with anoxygenic bacteria, such as *Desulfuromonas,* as well as chemoautotrophs associated with methanotrophy and methylotrophy (Appendix A). Due to the high-energy requirements of the N_2_ fixation reaction, it has been suggested that dissolved organic matter stimulates heterotrophic diazotrophs in aphotic environments [74,75,81], as well as in coastal waters [82,83]. Furthermore, the interior of carbon-rich particles has been described as a suitable microenvironment for chemosynthetic and heterotrophic N_2_ fixation [72,73,84,85]. This scenario could also explain the presence of *nif*H sequences closely affiliated with soil and marine sediments in the shallow waters of Chile Bay (maximum depth between 80–230 m), where water mixing conditions have been reported [53], which could enhance the suspension of particulate matter from bottom sediment. Another possible explanation for the presence of these sequences is that they originated from a nearby glacial source in the bay as a result of runoff due to ice melt [69].

The new N_2_ fixation rates detected in Chile Bay support the importance of de novo nitrogen fixed through this process in Western Antarctic coastal waters. These results demonstrate that during the austral summer, new inorganic nitrogen is actively produced, thereby suggesting that N_2_ fixation is necessary to supplement the inorganic nitrogen needs of the bacterial community growth in Western Antarctic coastal systems, as already reported for coastal upwelling regions at other latitudes [11,86,87,88]. Interestedly, nitrogen fixation rates in Chile Bay were remarkably high in the dark incubations (covered samples), pointing to the key role of heterotrophic and chemoautotrophic organisms, which is supported by our *nif*H gene analysis and has been suggested for other lighted temperate oceanic regions [28,89]. Given the high metabolic cost of the N_2_ fixation process, it is not surprising that diazotrophic photoautotrophs, such as Cyanobacteria, are most common in photic habitats of many other marine systems [2,3,28,88,90,91,92]. This is due to the significant advantage of diazotrophic photoautotrophs over their heterotrophic counterparts, whose N_2_ fixation capacity is limited by the availability of suitable organic carbon sources [93,94,95,96]. However, in this study, we did not detect any *nif*H gene OTUs related to photoautotrophs, such as cyanobacteria and the UCYN-A symbiont, as detected in other polar regions [11,97]. The high availability of organic carbon provided by eukaryotic photosynthesis and the dark chemoautotrophic activity observed in these waters of Chile Bay, which is reinforced by the possible occurrence of diazotrophic chemoautotrophs and heterotrophs, further support the cosmopolitan distribution of these organisms in the ocean [28,73,89], which now includes Western Antarctica. Chemoautotrophy, an important process for dark carbon fixation in Chile Bay, has previously been detected by active genes related to ribulose-1,5-bisphosphate carboxylase/oxygenase (RuBisCO) (the Calvin cycle) [39]; however, only <5.6% of all best-hit related sequences in our phylogenetic reconstruction of the *nif*H gene were identified as chemoautotrophs with potential as N_2_ fixers; that is, a coupling of both processes. This finding suggests that CO_2_ and N_2_ fixation may be uncoupled in most of the related genomes referenced here. However, more research is needed to corroborate the actual contribution of N_2_ fixation to the primary production in this system and also to better understand what physiological adaptations are required to carry out this process in an oxygenated environment, such as Chile Bay.

### 4.2. N_2_ Fixation Contribution to New Nitrogen in Chile Bay Seawater

Here we report the first detectable N_2_ fixation rates in Antarctic surface coastal waters of the WAP using light (uncovered) and dark (covered) incubations over several consecutive summers. This potential contribution to new nitrogen has yet to be well quantified in many regions, including Antarctica; however, doing so, with rates such as those reported here, would improve estimates of ocean nitrogen budgets [26].

As commented above and as previously reported in the literature, possible contamination with inorganic nitrogen occurred due to use of ^15^N_2_ gas batch SZ1670 from Sigma-Aldrich [54,56]. With no ability to correct the data, as reported in [55], this could reduce the validity and prevent comparison of rates obtained in this study between 2013–2014. However, the rates recorded from 2016–2019 were obtained using a stock of ^15^N_2_ gas from Cambridge Isotope Laboratories Inc., from which, to the best of our knowledge, no contamination has been reported to date. The rates recorded from 2016–2019 were always higher than those from years 2013 and 2014 and were also similar to those questioned rates [98,99] reported by [11] (44 nmol N L^−1^ d^−1^) around ice-covered regions in the Southern Ocean, but with the notable exception that the highest rates recovered here were under dark incubations. Our rates were also similar to those already reported for other illuminated open water areas of the Southern Ocean [32,33] and Arctic Ocean [10,31]. Furthermore, these N_2_ fixation rates (light and dark incubations) from Chile Bay were also as high as those obtained in illuminated tropical and subtropical waters [100,101], oligotrophic [3] and oxygen-deficient [76] oceanic waters. Notably, the higher N_2_ fixation rates detected at ≤30 m depth in Chile Bay under dark incubations highlight the relevance that this process may have in the water column through the potential activity of non-photosynthetic organisms. The rates obtained under both light and dark conditions in this high-latitude region suggest that N_2_ fixation is a relevant process for the incorporation of new nitrogen into this polar marine system during the summer. From these results, the question arises as to whether this N_2_ fixation in darkness will be as or more significant in these same waters during the continuous darkness of the polar winter. This is crucial, as N_2_ fixation in polar regions has been largely neglected in estimates of the global nitrogen budget. Therefore, these results represent novel relevant data to be considered in terms of the global nitrogen budget.

Although the diazotrophic contribution to marine production has long been investigated in relation to the global nitrogen budget [2,4,6,10,31,88,102], no estimates of this contribution have been attempted, thus far, in the coastal marine regions of the WAP, including under dark conditions. Based on the potential chemoautotrophic and heterotrophic N_2_ fixation capacity found here, as detected in the 2 m covered incubations during the summers from 2016–2019, we estimate that dark N_2_ fixation may contribute >37% to new nitrogen in the surface waters of Chile Bay. In contrast, a maximum of only 9% (2016) was observed for the uncovered light incubations.

Further extrapolation was conducted for the upper 30 m of the water column using the highest rates of dark N_2_ fixation reported in 2018. Assuming that Chile Bay is approximately 15 km^2^, we calculate that this process under the dark condition could fix ≥2.5 g N m^−2^ y^−1^, representing a significant potential contribution to the local nitrogen budget. Unfortunately, integration of the water column under the light condition was not possible due to the absence of this light incubation data set. This contribution should be taken with caution, as N_2_ fixation still needs to be investigated at deeper depths from 80–250 m (the maximum depths of station P3 and Chile Bay, respectively), as well as in other coastal areas of the WAP to obtain a more comprehensive integration of this process in the water column. Nevertheless, we can suggest that Chile Bay, and probably other coastal locations of the WAP, represent important nitrogen reservoirs facilitated by the process of dark N_2_ fixation by the heterotrophic community.

### 4.3. Effect of Environmental Variables on N_2_ Fixation Rates

Inorganic nitrogen compounds are known inhibitors of diazotrophic activity [5]. However, there is increasing evidence that biological N_2_ fixation may not be as sensitive to inorganic nitrogen as previously thought [16,103,104], especially when phosphorus is not limiting [15]. In Chile Bay, the highest N_2_ fixation rates (considering only the reliable rates between 2016–2019) coincided with high NO_3_^−^ (>20 µM) and NH_4_^+^ (>0.5 µM) concentrations, as well as with higher N:P ratios (Table 1). The average N:P ratio recorded in Chile Bay was 14:1, which is close to the Redfield ratio (16:1). The variations observed in the N:P ratio throughout the different summers reflect changes in the composition of the planktonic community that can differentially assimilate nutrients and export them from the surface to deeper waters. A high presence of diatoms has previously been associated with a low N:P ratio (~11:1) in the Southern Ocean [105], and is similar to what was observed in Chile Bay during the summers of 2013, 2014 and 2016, where a high presence and activity of Thalassiosirales and Bacillariales were reported [39,69,70]. This is a recurrent pattern observed in Antarctic coastal waters, such as the Ross Sea, where a low local N:P equilibrium was associated with diatom dominance, and a shift to a high N:P ratio was associated with Prymnesiophyte dominance, generally maintaining an N:P export ratio < 16:1 within a relatively small spatial scale [106]. This scenario suggests that the N:P ratio may play a critical role in Chile Bay, allowing N_2_ fixation to occur despite high concentrations of inorganic nitrogen, as long as phosphorus is available [15]. This trend was evident in Chile Bay, where phosphorus was the only factor (of those tested) that showed a negative correlation with both N_2_ fixation and carbon assimilation rates. Consequently, the limiting effect of the phosphorus concentration on nitrogen fixation under light conditions could be associated with the growth of diazotrophic photoautotrophs, while its effect under dark chemosynthesis could be associated with the growth of mostly heterotrophic organisms. However, more effort is needed to evaluate the real effect of phosphorus and other elements, such as iron (which was not evaluated in the present study), on diazotrophic organisms, like that seen for *Trichodesmium* cultures from the central Atlantic Ocean [107]. In addition, to better understand the functioning of these diazotrophic heterotrophs and chemoautotrophs found here in the WAP, laboratory studies on the physiological adaptations to avoid oxygen-induced nitrogenase inhibition [108,109,110] are now necessary, as are studies on the existence of potential microenvironments in sinking particles [73,85].

In summary, this study in the coastal marine region of the WAP provides novel information on N_2_ fixation, a relevant process of the nitrogen cycle. The high rates of N_2_ fixation detected in Chile Bay, mainly under dark conditions in both the surface layer and at 30 m, suggest that the energy requirements for primary production by chemosynthesizers could be sufficiently supported. Overall, these rates could contribute up to 37% of the new production in this nitrogen-based system, representing a contribution of ≥2.5 g N m^−2^ y^−1^. This study also reveals, for the first time in the WAP, the identity of specialized diazotrophs affiliated with heterotrophic and chemoautotrophic bacteria of nitrogenase clusters I and III and their role as major nitrogen fixers. This expands their ubiquity and potential biological contribution to Antarctic marine production, which has important implications for atmospheric CO_2_ sequestration.

These new findings should be considered in the context of global N_2_ fixation records, reinforcing the relevance of chemosynthetic and heterotrophic N_2_ fixers in the oceans, especially in polar regions where marine cyanobacteria are not particularly abundant. Ultimately, this will help to better understand the response of these marine systems to current and future global climate change scenarios.

## Figures and Tables

**Figure 1 microorganisms-10-01140-f001:**
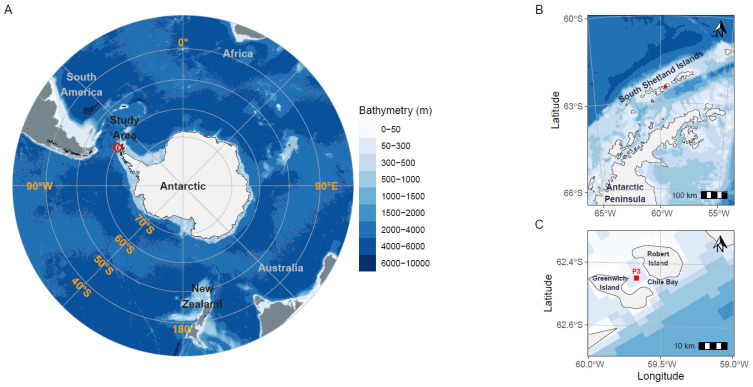
Geographic location of Chile Bay on Greenwich Island, Western Antarctic Peninsula (WAP). The red square (**A**–**C**) represents the P3 sampling point in the bay where seawater was collected to measure environmental variables and N_2_ fixation and carbon assimilation rates.

**Figure 2 microorganisms-10-01140-f002:**
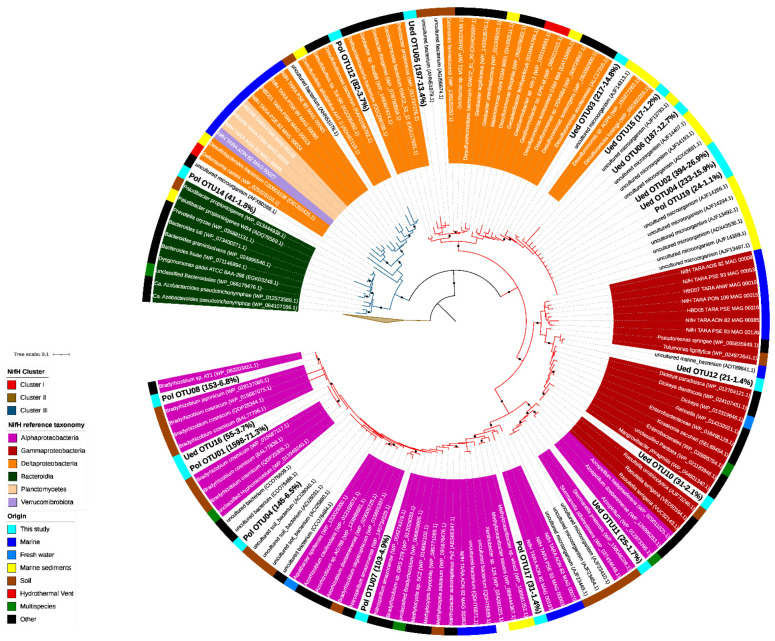
Phylogenetic placement of NifH protein sequences identified in coastal waters of Chile Bay. The phylogenetic placement was carried out using a maximum-likelihood phylogenetic reconstruction that included annotated reference NifH protein sequences and novel NifH protein sequences retrieved from Tara Oceans metagenomes. Short NifH amino acid sequences are shown with a white background. NifH sequences obtained from Chile Bay are shown in bold. The number of reads assigned to each OTU, and their respective relative abundance, is shown in parenthesis (# of reads-% relative abundance). The tree was rooted to archaeal NifH sequences from Cluster II. The branch color indicates the NifH cluster, while the label background indicates the taxonomy of the NifH sequence. Black circles in the nodes indicate ≥95% of ultra-fast bootstrap and ≥80% of SH-aLRT branch support.

**Figure 3 microorganisms-10-01140-f003:**
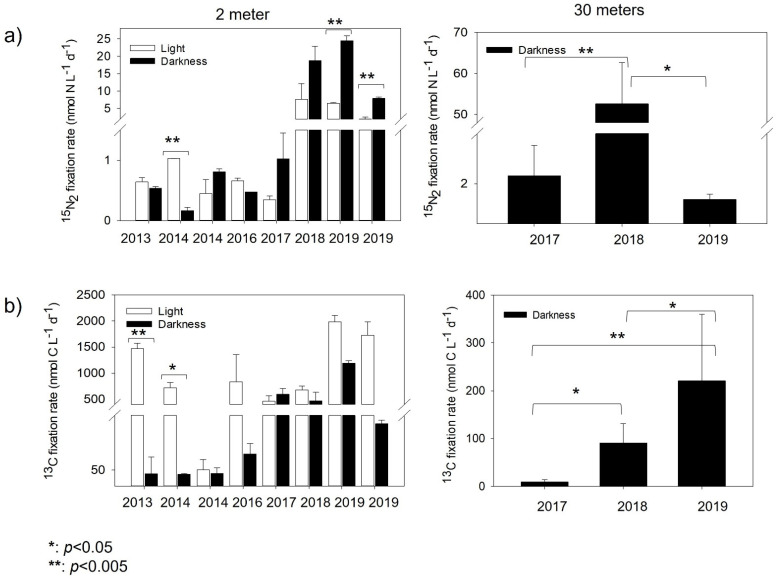
Interannual marine nitrogen ((**a**), ^15^N_2_) fixation and carbon ((**b**), ^13^C) assimilation rates in Chile Bay during the late summer in 2013–2014 and 2016–2019. The white bars represent light incubations and black bars represent dark incubations. The error bars represent triplicate rates. Significant differences between rates (*p-*value) were evaluated by the *t*-test.

**Figure 4 microorganisms-10-01140-f004:**
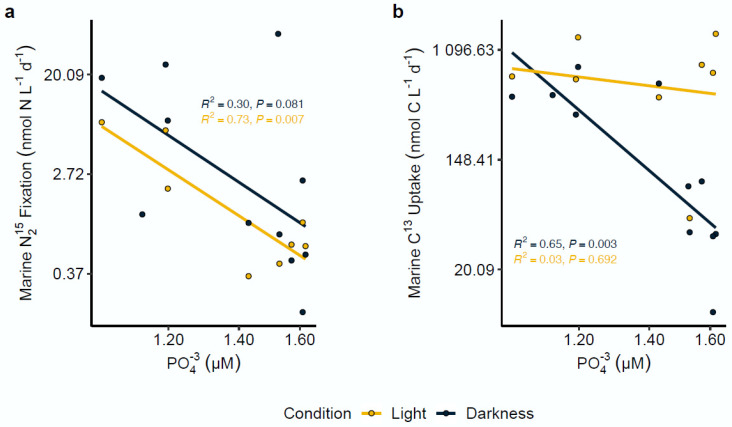
Nitrogen fixation and carbon assimilation rates in association with phosphate (PO_4_^3−^) concentrations. Light and dark nitrogen fixation (**a**) and carbon assimilation (**b**) were strongly correlated with PO_4_^3−^ concentrations in Chile Bay. All axes were transformed to log scale. R^2^ and *p*-value significance values for each analysis are shown on each plot.

**Table 1 microorganisms-10-01140-t001:** Chile Bay physicochemical variables during the late summer periods (February–March) of 2013–2014 and 2016–2019. Inorganic nutrients (NO_3_^−^, NO_2_^−^ NH_4_^+^, PO_4_^3−^) and the N:P ratio [N = (NO_3_^−^ + NO_2_^−^ + NH_4_^+^); *p* = PO_4_^3−^] were assessed.

Date	Depth (m)	Temperature °C	Salinity	Chl-*a* (mg L^−1^)	NH_4_^+^ (µM)	NO_2_^−^ (µM)	NO_3_^−^ (µM)	PO_4_^3−^ (µM)	N:P
8 February 2013	2	0.55	34.15	0.03	NaN	0.32	20.52	1.62	12.85
14 February 2014	2	−0.11	33.20	0.36	NaN	0.16	19.52	1.61	12.22
22 February 2014	2	0.34	33.47	1.18	NaN	0.13	17.16	1.53	11.30
22 February 2016	2	−0.25	NaN	1.01	NaN	0.29 ± 0.03	14.56 ± 0.16	1.57 ± 0.16	9.45
10 February 2017	2	2.23	34.03	0.69 ± 0.32	0.33 ± 0.07	0.22 ± 0.02	21.64 ± 2.12	1.43 ± 0.11	15.52
21 February 2017	30	1.42	34.16	0.327 ± 0.17	0.61 ± 0.09	0.204 ± 0.03	23.3 ± 4.55	1.61 ± 0.13	14.98
9 February 2018	30	1.33	34.05	1.55 ± 1.63	0.96 ± 0.29	0.26 ± 0.01	20.90 ± 3.73	1.53 ± 0.30	14.50
16 February 2018	2	1.61	34.01	3.58 ± 2.07	0.93 ± 0.09	0.26 ± 0.04	17.70 ± 5.50	1.04 ± 0.42	18.22
17 February 2019	2	1.54	34.10	2.14 ± 1.90	0.48 ± 0.18	0.17 ± 0.03	16.50 ± 3.97	1.19 ± 0.27	14.39
8 March 2019	2	1.11	34.36	4.17 ± 2.03	1.19 ± 0.54	0.16 ± 0.04	15.55 ± 4.31	1.20 ± 0.23	14.11
8 March 2019	30	1.16	34.21	2.83 ± 1.81	1.27 ± 0.78	0.17 ± 0.04	17.28 ± 4.06	1.13 ± 0.26	16.52

± = standard deviation.

**Table 2 microorganisms-10-01140-t002:** Contribution of marine N_2_ fixation to local primary production (N_2_LPP) and new production (N_2_NP) in Chile Bay based on the C:N ratio and nitrogen (N_2_) and carbon (C) fixation rates under light and dark conditions.

						^15^N_2_ Contribution to	
Date	Sample Depth (m)	Condition	molC:molN	Marine ^15^N_2_ Fixation	Total Marine ^13^C Uptake	Total N_2_LPP (C) *	Total N_2_P (N) **
				(nmol N L^−1^ d^−1^)	(nmol C L^−1^ d^−1^)	%	%
8 February 2013	2	Light	10.49	0.64 ± 0.06	1471 ± 101.51	0.46	0.76
14 February 2014	2	Light	7.47	1.03 ± 0.002	723 ± 97.91	1.07	4.29
22 February 2014	2	Light	5.95	0.45 ± 0.14	51.18 ± 31.50	5.23	0.66
22 February 2016	2	Light	6.80	0.66 ± 0.05	837 ± 5515.2	0.54	9.82
10 February 2017	2	Light	5.82	0.35 ± 0.09	463 ± 104	0.44	0.31
16 February 2018	2	Light	5.02	7.70 ± 4.42	677 ± 76.3	5.71	3.84
17 February 2019	2	Light	4.11	6.54 ± 0.20	643	4.18	4.35
8 March 2019	2	Light	3.11	2.03 ± 0.57	1380	0.46	1.39
8 February 2013	2	Darkness	0.18	0.54 ± 0.03	38.37 ± 22.04	0.25	4.66
14 February 2014	2	Darkness	7.65	0.17 ± 0.05	36.80 ± 2.93	3.50	0.60
22 February 2014	2	Darkness	6.43	0.81 ± 0.05	39.60 ± 17.53	13.15	0.82
22 February 2016	2	Darkness	6.31	0.48	100 ± 32.51	3.00	37.59
10 February 2017	2	Darkness	4.72	1.02 ± 0.73	595 ± 109	0.81	2.26
16 February 2018	2	Darkness	4.22	18.78 ± 2.19	468 ± 165	16.93	5.76
17 February 2019	2	Darkness	4.69	24.51 ± 1.40	338 ± 167	34.01	36.96
8 March 2019	2	Darkness	3.29	7.96 ± 0.38	805 ± 345	3.25	15.99
21 February 2017	30	Darkness	4.96	2.39 ± 1.53	9.24 ± 4.70	128	5.13
9 February 2018	30	Darkness	7.10	45.40 ± 10.09	91.47 ± 40.09	354	12.88
8 March 2019	30	Darkness	3.55	1.21 ± 0.27	482 ± 123	0.89	2.80

* N_2_LPP = contibution of N_2_ fixation on local primary production based on carbon; ** N_2_P= contibution of N_2_ fixation on new production based on nitrogen; ± =standard deviation.

## Data Availability

The required data set is already available in manuscript and the Appendix A.

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
