# Peer review of "Dark Diazotrophy during the Late Summer in Surface Waters of Chile Bay, West Antarctic Peninsula"

_microorganisms, 2022, doi:10.3390/microorganisms10061140_

Round 1

Reviewer 1 Report

Alcamán-Arias et al. examine N2 and C fixation rates, as well as nitrate assimilation rates, along with diazotroph community composition in Chile Bay, off the coast of Greenwich Island in the Antarctic. They use the measured rates to estimate the percent contribution of N2 fixation to local primary production and total new production, which is surprisingly quite high for this region where nitrate-fueled PP would be expected. Furthermore, they speculate that putative heterotrophic diazotrophs are the main N2 fixers in the system.

Unfortunately, it is impossible to evaluate the accuracy of the N2 fixation rate measurements – which are fundamental to this study and the implications of their findings. They utilize a 15N2 tracer gas from a company (Sigma-Aldrich) that is well documented to have 15N-ammonium, 15N-nitrate/nitrite and 15N-nitrous oxide that can grossly skew the 15N of particulate N due to inorganic N uptake in by non-N2 fixers in these incubations, and can result in false positive N2 fixation rates. It is absolutely critical to include data on the inorganic N contamination on the specific lots of 15N2 gas used in this study, prior to considering this for publication. I acknowledge this is not always possible, but at a bare minimum the authors need to consult Dabundo et al., (2014) and White et al., (2020), and make estimates of the range of potential contribution from potential 15N contamination.           

Dabundo, R., Lehmann, M.F., Treibergs, L., Tobias, C.R., Altabet, M.A., Moisander, P.H. and Granger, J., 2014. The contamination of commercial 15N2 gas stocks with 15N–labeled nitrate and ammonium and consequences for nitrogen fixation measurements. PloS one9(10), p.e110335.

White, A.E., Granger, J., Selden, C., Gradoville, M.R., Potts, L., Bourbonnais, A., Fulweiler, R.W., Knapp, A.N., Mohr, W., Moisander, P.H. and Tobias, C.R., 2020. A critical review of the 15N2 tracer method to measure diazotrophic production in pelagic ecosystems. Limnology and Oceanography: Methods18(4), pp.129-147.

Author Response

R: We greatly thank all the reviewers for their comments and criticisms regarding the possible contamination of the Sigma N2 gas vials, and apologize for the errors made in this manuscript.

Trying to understand the magnitude of the error in the measured rates, we corroborate that indeed for the measurements performed in this study between the years 2013 and 2014, Sigma-Aldrich brand N2 gas batch SZ1670 was used. Therefore, these rates must indeed be taken with great caution, due to possible contamination. However, we verified that the vials of N2 gas used from 2016 to 2019 were fortunately purchased from Cambridge Isotope Laboratories, Inc. whose manufacturing has shown no reported contamination to date. These have now been clarified, and new information was added to the manuscript. Thus, the rates recorded by our work since 2016 are under the expected standards in the use of 15N2 gas stocks. New information was added in the methods section and in discussion about this possible error due to contamination in the rates obtained during the first years of study (2013-2014) when using the 15N2 Sigma batch, reported by Dabundo et al., 2014 and Böttjer et al., 2017. 

These rates from 2016 were now the only ones actually discussed in this study, and were similar to those previously reported (44 nmol N L-1 d-1; (reference 11)) around ice-covered regions in the Southern Ocean, with the exception that the highest rates recovered here were under dark conditions. Our rates were also similar to those already reported for other illuminated open-water areas of the Southern Ocean (33,34) and Arctic Ocean (10,31), as now described in the manuscript.

Unfortunately, we cannot make the corrections that the referee requests because we no longer have the original N2 gas batches used for the years in question. For this reason we cannot, for example, test the gas dilution curves in water or evaluate possible contamination of the standards of reference used in the IRMS, among other corrections mentioned in the final work of White et al., 2020.

Reviewer 2 Report

I really enjoyed reading “Light and dark diazotrophy during the summer in surface waters of Chile Bay, West Antarctic Peninsula.” by Alcamán-Arias. This is one of the most exciting studies I read recently, and I feel that there is a strong potential to shift the paradigm of N2 fixation in a global sense. A recent study (Shiozaki et al., 2020) shows a high rate of N2 fixation in Antarctic Ocean as the authors point out. This study further solidifies the potential of N2 fixation in the coastal Antarctic Ocean. I am a mathematical/computational/quantitative modeler (thus not a observationalists), and it is not easy to judge the methodology, but given that high rates of N2 fixation is observed in many points, I feel that the results are solid. I think the texts are well written and the figures are well presented. However, there are places that the manuscript may further improve to maximize the impact of these great findings. Below I describe these in detail. Overall, I would like to congratulate the author for such a great discovery and wish them the best for further investigations.

Major comments/suggestions:

In “however, these estimates are variable across the Arctic Ocean (11–13).” I think the author should be citing “Shiozaki, T. et al. Diazotroph community structure and the role of nitrogen fixation in the nitrogen cycle in the Chukchi Sea (western Arctic Ocean). Limnol. Oceanogr. 63, 2191–2205 (2018).” Reference (11) is about Antarctic ocean not Arctic. However, the reference (11) below is appropriate when the authors talk about the Antarctic ocean.

Figure 1 is great. At the same time, not so many people in the world are intuitive about the location when it comes to the details in the Antarctic. To reach a broader audience, I suggest adding a map of the whole Antarctic (and maybe including some part of other continents like South America and Australia) so that the readers can get more intuition about the location. Also, in N2 fixation, latitudes are important, as a study like this pushes the latitudinal limit of the N2 fixation. Therefore, I recommend adding latitudinal information to the map. Great figure though.

Because this paper shows that the heterotrophic bacteria are the good candidate for the major contributor of N2 fixation, I recommend discussing the potential mechanisms citing “Chakraborty, S., K. H. Andersen, A. W. Visser, K. Inomura, M. J. Follows, and L. Riemann. 2021. Quantifying nitrogen fixation by heterotrophic bacteria in sinking marine particles. Nat. Commun. 12: 4085. doi:10.1038/s41467-021-23875-6”in the discussion. This paper explores the mechanisms of how N2 fixation may occur under the high availability of reactive N and O2. The authors may argue that the dark condition favors N2 fixation due to lower O2 in the environment.

LPP and NP equations should be located near the center.

Is the transcriptomics performed? In Shiozaki et al (2020), UCYN-A was small in DNA but it turned out to be the major contributor in RNA. If not, it might be worth discussing the possibility of UCYN-A’s contribution?

It is not easy to get an intuition of “nitrogen-based new production (NP).” Is this the contribution based on the N2 fixation? If so, it should be clarified. For example, it can be named as “dinitrogen-based new production (N2P)”(with 2 in subscript). By looking at the equation, NP sounds more like the fraction of N uptake based on the N2 fixation. It could be more intuitive to define it as “fraction of N2 fixation for N assimilation”. These are just my suggestions for potential improvement, so I leave the authors for the choice.

Also, LPP is defined as local primary production. However, looking at the equation, it looks more like a contribution from N2 fixation. Thus, I would clarify that in the definition. For example, it could be defined as “contribution of N2 fixation for the local primary production (N2LPP) with 2 in subscript.

Similarly, NP could be defined as N2 based new production (N2P). By simply adding “2” it becomes more intuitive as it seems largely for N2 fixation, the main topic of this paper. Also, NP and N:P look similar so adding “2” (subscript) would avoid confusion.

Regarding Figure 5, would it be possible to add panels for NO3-/PO43- for the x axis? I think people are interested in the rate of N2 fixation in relation to N:P of the nutrient. This information seems there in table one so I think I could be done simply?

Regarding “However, regions of particular importance for C sequestration, such as the coastal areas of Antarctica (63–65), have so far been totally ignored with respect to estimating N2 fixation rates,” Deutsch et al 2007 (445,163-167 Nature) shows a high rate of N2 fixation in Antarctica, so it may not be completely ignored. I would say “largely ignored” instead of “totally ignored”. I agree with the author in that Antarctica got little attention, but it requires much more attention.

Regarding “Specially, revealing the identity, distribution and contribution of diazotrophs in Antarctic coastal seawater has been neglected.” Shiozaki et al 2020 made an effort to reveal this. Thus, I would rephrase that to say “Especially, revealing the identity, distribution and contribution of diazotrophs in Antarctic coastal seawater have only recently started (Shiozaki et al 2020)”. I do agree that this present study has a very high value and it is certainly the first in the WAP.

Regarding “However, and given the high metabolic cost of the N2 fixation process, it is not surprising that in photic habitats of many other marine systems, diazo-trophic photoautotrophs, such as Cyanobacteria, are the most common diazotrophs (2,3,28,74,81–83), which is due to the significant advantage over their heterotrophic counterparts, whose N2 fixation capacity is limited by the availability of suitable organic carbon sources (84,85).” I agree with the advantage. However, since nitrogenase is sensitive to O2, daytime may be a disadvantage, since the byproduct of photosynthesis is O2. I think the author can discuss the negative effect of O2 on N2 fixation. This has been explored in recent modeling studies, some of which authors may consider citing.

Inomura K, Wilson ST, Deutsch C (2019) Mechanistic model for the coexistence of nitrogen fixation and photosynthesis in marine Trichodesmium. mSystems 4:e00210-19.

Inomura K, Deutsch C, Wilson ST, Masuda T, Lawrenz E, Sobotka R, Bučinská L, Gauglitz JM, Saito MA, Prášil O, Follows MJ (2019) Quantifying oxygen management and temperature and light dependencies of nitrogen fixation by Crocosphaera watsonii. mSphere 4:e00531-19.

Inomura K, Deutsch C, Masuda T, Prášil O, Follows MJ (2020) Quantitative models of nitrogen-fixing organisms. Computational and Structural Biotechnology Journal 18:3905–3924.

At the same time, I agree with the potential advantage of photosynthesis as it provides C. A modeling study shows that the ratio of resource C:N is important in the occurrence of N2 fixation. The author may cite the following paper to support their argument:

Inomura K, Bragg J, Follows MJ (2017) A quantitative analysis of the direct and indirect costs of nitrogen fixation: a model based on Azotobacter vinelandii. ISME Journal 11:166–175.

I am very intrigued by the results that there is higher N2 fixation during the dark period. I suggest discussing the cause of this result more deeply. The above listed references would be useful in such a discussion, as they quantitatively explore the effect of  C and O2 and available reactive N, the availability of which changes between the dark and light periods.

In the paragraph that starts with “Finally, analysis of the nifH gene enabled us to reveal...” I feel that discussing how N2 fixation is conducted would deepen the argument. Since this paragraph talks about the heterotrophic N2 fixation in particles, the author may very briefly talk about how N2 fixation was made possible in sinking particles citing “Chakraborty, S., K. H. Andersen, A. W. Visser, K. Inomura, M. J. Follows, and L. Riemann. 2021. Quantifying nitrogen fixation by heterotrophic bacteria in sinking marine particles. Nat. Commun. 12: 4085. doi:10.1038/s41467-021-23875-6”.

Re: “These new findings should be considered in the context of global N2 fixation records, reinforcing the relevance of heterotrophic N2 fixers in the oceans, especially in polar regions where marine cyanobacteria are not particularly abundant. Ultimately, this will help to better understand the response of these marine systems to current and future global climate change scenarios.” I think this is a great conclusion and I congratulate the authors again for such a great discovery.

Minor comments/suggestions:

Page 2:

“as well as with identification of the main factors” -> “as well as with the identification of the main factors”

“Despite this, neither nitrogen fixation rates nor the occurrence of diazotrophs have yet been reported for the West Antarctic” -> “Despite this, neither nitrogen fixation rates nor the occurrence of diazotrophs has yet been reported for the West Antarctic”

“limiting our knowledge for the actual geographic distribution” -> “limiting our knowledge of the actual geographic distribution”

Page4:

Re (Regarding): “To estimate carbon (H13CO3-) uptake, the same bottles were amended with 2.5-mL of H13CO3- (3.645 mg 13C mL-1),” In this sentence, the subject is “the same bottles” but they do not estimate carbon. The subject of “To estimate carbon XXX” should match the subject of the sentence, which should be people conducting the experiments. Thus, I suggest modifying it to “To estimate carbon (H13CO3-) uptake, we amended the same bottles with 2.5-mL of H13CO3- (3.645 mg 13C mL-1),”

Re: “mgC:mgN (C:N) ratio” mg is fine but it might be more intuitive to use mol/mol as in Redfield ratio.

Page 5:

Re: “MAFFT (--keeplength --add)” I wonder if this is correct? If so, that is fine.

Re: “To determine potential drivers modulating late summer N2 fixation and carbon assimilation activities in Chile Bay, linear regressions were performed between the obtained N2 and C rates and the biogeochemical variables of temperature, Chl-a, NO3-, NO2-, NH4+ and PO43-.” For the same reason as above, the subject of the main clause of the sentence should be people who conducted the experiment. Thus I suggest as follows “To determine potential drivers modulating late summer N2 fixation and carbon assimilation activities in Chile Bay, we performed linear regressions between the obtained N2 and C fixation rates and the biogeochemical variables of temperature, Chl-a, NO3-, NO2-, NH4+ and PO43-.”

“homogenously” -> “homogeneously”

Page 6:

“within the same year” -> “within the same year,”

Page 7:

Table 2, please add the unit of C:N. It is in mg/mg in the text, but is it mol/mol here? It would be good to clarify it.

Page 8:

“The error bars represents triplicate rates.” -> “The error bars represent triplicate rates.”

Page 9:

“Figure 3. Nitrogen fixation and carbon assimilation rates association with phosphate (PO43-) concentrations.” -> Figure 3. Nitrogen fixation and carbon assimilation rates in association with phosphate (PO43-) concentrations.

“Light and dark nitrogen fixation (a) and carbon assimilation (b) was strongly related with PO43- concentrations in Chile Bay.” -> “Light and dark nitrogen fixation (a) and carbon assimilation (b) were strongly correlated with PO43- concentrations in Chile Bay.”

“R2 and p-value significance values for each analysis are showed on each plot.” -> “R2 and p-value significance values for each analysis are shown on each plot.”

In Figure 3, I think “-3” on PO4 should be “3-” as in other places.

Page 10:

Re: “NifH sequences obtained from Chile Bay are shown in bold (Pol=PolF/PolR, Ued=Ueda19F/R6).” Maybe it should be nifH as it talks about genes? If it is NifH, it should indicate proteins.  I am not a specialist in omics, so please check it.

Page 11:

“the under sampling” -> “the undersampling”

“in this study” -> “in this study,”

“For that, over six late summer periods” -> “For that, over six late summer periods,”

“These new rates recorded in Chile Bay are similar as those reported by (11) (44 nmol N L-1 d-1) with the exception that the higher rates recovered here were under dark conditions.” -> “These new rates recorded in Chile Bay are similar to those reported by (11) (44 nmol N L-1 d-1), with the exception that the higher rates recovered here were under dark conditions.” (here “as” should be “to” and “,” before “with the exception” would help the reader.

Page 12:

“nitrogen, facilitated by the process of biological N2 fixation.” Please, remove “,” after nitrogen. Otherwise, the “facilitated” may modify the subject of the main clause.

“we conducted a study in Chile Bay that, for the first time, reveals new knowledge about diazotrophy in the coastal seawaters of the WAP.” -> “we conducted a study in Chile Bay that, for the first time, revealed new knowledge about diazotrophy in the coastal seawaters of the WAP.”

“In this bay” -> “In this bay,”

Figures:

The resolution is not as high as it could be. Maybe it is just the version I got. If there is room for increasing the resolution, I suggest increasing the resolution to maximize the impact of this great paper.

Author Response

I really enjoyed reading “Light and dark diazotrophy during the summer in surface waters of Chile Bay, West Antarctic Peninsula.” by Alcamán-Arias. This is one of the most exciting studies I read recently, and I feel that there is a strong potential to shift the paradigm of N2 fixation in a global sense. A recent study (Shiozaki et al., 2020) shows a high rate of N2 fixation in Antarctic Ocean as the authors point out. This study further solidifies the potential of N2 fixation in the coastal Antarctic Ocean. I am a mathematical/computational/quantitative modeler (thus not a observationalists), and it is not easy to judge the methodology, but given that high rates of N2 fixation is observed in many points, I feel that the results are solid. I think the texts are well written and the figures are well presented. However, there are places that the manuscript may further improve to maximize the impact of these great findings. Below I describe these in detail. Overall, I would like to congratulate the author for such a great discovery and wish them the best for further investigations.

 R: We greatly appreciate the reviewer's positive comments and constructive criticism, thanks to which we were able to improve the manuscript.

Major comments/suggestions:

In “however, these estimates are variable across the Arctic Ocean (11–13).” I think the author should be citing “Shiozaki, T. et al. Diazotroph community structure and the role of nitrogen fixation in the nitrogen cycle in the Chukchi Sea (western Arctic Ocean). Limnol. Oceanogr. 63, 2191–2205 (2018).” Reference (11) is about Antarctic ocean not Arctic. However, the reference (11) below is appropriate when the authors talk about the Antarctic ocean.

R: The reviewer is right. It was entirely our mistake not to have incorporated this reference in the original manuscript. Now the reference has been included. 

Figure 1 is great. At the same time, not so many people in the world are intuitive about the location when it comes to the details in the Antarctic. To reach a broader audience, I suggest adding a map of the whole Antarctic (and maybe including some part of other continents like South America and Australia) so that the readers can get more intuition about the location. Also, in N2 fixation, latitudes are important, as a study like this pushes the latitudinal limit of the N2 fixation. Therefore, I recommend adding latitudinal information to the map. Great figure though.

R: Thanks for the recommendation. We have changed the figure with a new perspective, and added more details of the Antarctic location of this study.

Because this paper shows that the heterotrophic bacteria are the good candidate for the major contributor of N2 fixation, I recommend discussing the potential mechanisms citing “Chakraborty, S., K. H. Andersen, A. W. Visser, K. Inomura, M. J. Follows, and L. Riemann. 2021. Quantifying nitrogen fixation by heterotrophic bacteria in sinking marine particles. Nat. Commun. 12: 4085. doi:10.1038/s41467-021-23875-6”in the discussion. This paper explores the mechanisms of how N2 fixation may occur under the high availability of reactive N and O2. The authors may argue that the dark condition favors N2 fixation due to lower O2 in the environment.

R: The referee is right. Now we added this new citation proportionated by the referee. In addition, we discussed the potential anoxic microenvironment inside sinking particles according to the presence of  nifH sequences found in the Chile Bay oxygenated waters.

LPP and NP equations should be located near the center.

R: Changed.

Is the transcriptomics performed? In Shiozaki et al (2020), UCYN-A was small in DNA but it turned out to be the major contributor in RNA. If not, it might be worth discussing the possibility of UCYN-A’s contribution?

R: Yes, we check some metranscriptomics data from 2016 (Alcamán-Arias et al 2021) to search for UCYN-A presence, but we don’t find any signal associated to this cyanobacteria.  Moreover, we performed specific cDNA nifH gene sequencing but unfortunately, we did not obtained good results and the data was not finally included in the ms.

It is not easy to get an intuition of “nitrogen-based new production (NP).” Is this the contribution based on the N2 fixation? If so, it should be clarified. For example, it can be named as “dinitrogen-based new production (N2P)”(with 2 in subscript). By looking at the equation, NP sounds more like the fraction of N uptake based on the N2 fixation. It could be more intuitive to define it as “fraction of N2 fixation for N assimilation”. These are just my suggestions for potential improvement, so I leave the authors for the choice.

R: Yes, NP refers to the nitrogen (N) contribution from N2 fixation to new production (NP). For better understanding, we have changed these terms throughout the text for clarity.

Also, LPP is defined as local primary production. However, looking at the equation, it looks more like a contribution from N2 fixation. Thus, I would clarify that in the definition. For example, it could be defined as “contribution of N2 fixation for the local primary production (N2LPP) with 2 in subscript.

R: The suggestion was considered, and changes were made throughout the text.

Similarly, NP could be defined as N2 based new production (N2P). By simply adding “2” it becomes more intuitive as it seems largely for N2 fixation, the main topic of this paper. Also, NP and N:P look similar so adding “2” (subscript) would avoid confusion.

R: The suggestion was considered, and changes were made throughout the text.

Regarding Figure 5, would it be possible to add panels for NO3-/PO43- for the x axis? I think people are interested in the rate of N2 fixation in relation to N:P of the nutrient. This information seems there in table one so I think I could be done simply?

R: We don’t have Figure 5, maybe the referee is referring to Figure 3? We modified all the figures and Tables.

 Regarding “However, regions of particular importance for C sequestration, such as the coastal areas of Antarctica (63–65), have so far been totally ignored with respect to estimating N2 fixation rates,” Deutsch et al 2007 (445,163-167 Nature) shows a high rate of N2 fixation in Antarctica, so it may not be completely ignored. I would say “largely ignored” instead of “totally ignored”. I agree with the author in that Antarctica got little attention, but it requires much more attention.

R: Agree, sentence changed.

Regarding “Specially, revealing the identity, distribution and contribution of diazotrophs in Antarctic coastal seawater has been neglected.” Shiozaki et al 2020 made an effort to reveal this. Thus, I would rephrase that to say “Especially, revealing the identity, distribution and contribution of diazotrophs in Antarctic coastal seawater have only recently started (Shiozaki et al 2020)”. I do agree that this present study has a very high value and it is certainly the first in the WAP.

R: Changed.

 Regarding “However, and given the high metabolic cost of the N2 fixation process, it is not surprising that in photic habitats of many other marine systems, diazo-trophic photoautotrophs, such as Cyanobacteria, are the most common diazotrophs (2,3,28,74,81–83), which is due to the significant advantage over their heterotrophic counterparts, whose N2 fixation capacity is limited by the availability of suitable organic carbon sources (84,85).” I agree with the advantage. However, since nitrogenase is sensitive to O2, daytime may be a disadvantage, since the byproduct of photosynthesis is O2. I think the author can discuss the negative effect of O2 on N2 fixation. This has been explored in recent modeling studies, some of which authors may consider citing.

 Inomura K, Wilson ST, Deutsch C (2019) Mechanistic model for the coexistence of nitrogen fixation and photosynthesis in marine Trichodesmium. mSystems 4:e00210-19.

 Inomura K, Deutsch C, Wilson ST, Masuda T, Lawrenz E, Sobotka R, Bučinská L, Gauglitz JM, Saito MA, Prášil O, Follows MJ (2019) Quantifying oxygen management and temperature and light dependencies of nitrogen fixation by Crocosphaera watsonii. mSphere 4:e00531-19.

 Inomura K, Deutsch C, Masuda T, Prášil O, Follows MJ (2020) Quantitative models of nitrogen-fixing organisms. Computational and Structural Biotechnology Journal 18:3905–3924.

R: Thank you for the suggested references.

In this manuscript we decided not to discuss in depth the negative effect of O2 on N2 fixation, because we did not recruit any nifH sequence from phototrophic diazotrophs. Instead, we believe it is more interesting to emphasize more on the presence of a potential chemosynthetic fixation, so all these references have been added in the discussion section.

 At the same time, I agree with the potential advantage of photosynthesis as it provides C. A modeling study shows that the ratio of resource C:N is important in the occurrence of N2 fixation. The author may cite the following paper to support their argument:

 Inomura K, Bragg J, Follows MJ (2017) A quantitative analysis of the direct and indirect costs of nitrogen fixation: a model based on Azotobacter vinelandii. ISME Journal 11:166–175.

R: The reference was added. Many thanks.

I am very intrigued by the results that there is higher N2 fixation during the dark period. I suggest discussing the cause of this result more deeply. The above listed references would be useful in such a discussion, as they quantitatively explore the effect of  C and O2 and available reactive N, the availability of which changes between the dark and light periods.

R: We have modified the discussion section in order to clarify the meaning of our results.  

In the paragraph that starts with “Finally, analysis of the nifH gene enabled us to reveal...” I feel that discussing how N2 fixation is conducted would deepen the argument. Since this paragraph talks about the heterotrophic N2 fixation in particles, the author may very briefly talk about how N2 fixation was made possible in sinking particles citing “Chakraborty, S., K. H. Andersen, A. W. Visser, K. Inomura, M. J. Follows, and L. Riemann. 2021. Quantifying nitrogen fixation by heterotrophic bacteria in sinking marine particles. Nat. Commun. 12: 4085. doi:10.1038/s41467-021-23875-6”.

R: In the modified discussion we added this reference

Re: “These new findings should be considered in the context of global N2 fixation records, reinforcing the relevance of heterotrophic N2 fixers in the oceans, especially in polar regions where marine cyanobacteria are not particularly abundant. Ultimately, this will help to better understand the response of these marine systems to current and future global climate change scenarios.” I think this is a great conclusion and I congratulate the authors again for such a great discovery.

R: Many thanks

 Minor comments/suggestions:

Page 2:

“as well as with identification of the main factors” -> “as well as with the identification of the main factors”

 R: Changed

“Despite this, neither nitrogen fixation rates nor the occurrence of diazotrophs have yet been reported for the West Antarctic” -> “Despite this, neither nitrogen fixation rates nor the occurrence of diazotrophs has yet been reported for the West Antarctic”

 R: Changed

“limiting our knowledge for the actual geographic distribution” -> “limiting our knowledge of the actual geographic distribution”

R: Changed

Page4:

Re (Regarding): “To estimate carbon (H13CO3-) uptake, the same bottles were amended with 2.5-mL of H13CO3- (3.645 mg 13C mL-1),” In this sentence, the subject is “the same bottles” but they do not estimate carbon. The subject of “To estimate carbon XXX” should match the subject of the sentence, which should be people conducting the experiments. Thus, I suggest modifying it to “To estimate carbon (H13CO3-) uptake, we amended the same bottles with 2.5-mL of H13CO3- (3.645 mg 13C mL-1),”

R: Changed

Re: “mgC:mgN (C:N) ratio” mg is fine but it might be more intuitive to use mol/mol as in Redfield ratio.

R: Changed

Page 5:

Re: “MAFFT (--keeplength --add)” I wonder if this is correct? If so, that is fine.

R: It is correct. These are the options used to align the query NifH sequences to the reference MSA for the phylogenetic placement using the software MAFFT.

Re: “To determine potential drivers modulating late summer N2 fixation and carbon assimilation activities in Chile Bay, linear regressions were performed between the obtained N2 and C rates and the biogeochemical variables of temperature, Chl-a, NO3-, NO2-, NH4+ and PO43-.” For the same reason as above, the subject of the main clause of the sentence should be people who conducted the experiment. Thus I suggest as follows “To determine potential drivers modulating late summer N2 fixation and carbon assimilation activities in Chile Bay, we performed linear regressions between the obtained N2 and C fixation rates and the biogeochemical variables of temperature, Chl-a, NO3-, NO2-, NH4+ and PO43-.”

R: Modified

“homogenously” -> “homogeneously”

R: Modified

Page 6:

“within the same year” -> “within the same year,”

R: Modified

Page 7:

Table 2, please add the unit of C:N. It is in mg/mg in the text, but is it mol/mol here? It would be good to clarify it.

R: Modified to molC:molN

Page 8:

“The error bars represents triplicate rates.” -> “The error bars represent triplicate rates.”

  R: Modified

Page 9:

“Figure 3. Nitrogen fixation and carbon assimilation rates association with phosphate (PO43-) concentrations.” -> Figure 3. Nitrogen fixation and carbon assimilation rates in association with phosphate (PO43-) concentrations.

 R: Modified

“Light and dark nitrogen fixation (a) and carbon assimilation (b) was strongly related with PO43- concentrations in Chile Bay.” -> “Light and dark nitrogen fixation (a) and carbon assimilation (b) were strongly correlated with PO43- concentrations in Chile Bay.”

R: Modified

 “R2 and p-value significance values for each analysis are showed on each plot.” -> “R2 and p-value significance values for each analysis are shown on each plot.”

 R: Modified

In Figure 3, I think “-3” on PO4 should be “3-” as in other places.

R: Modified

 Page 10:

Re: “NifH sequences obtained from Chile Bay are shown in bold (Pol=PolF/PolR, Ued=Ueda19F/R6).” Maybe it should be nifH as it talks about genes? If it is NifH, it should indicate proteins.  I am not a specialist in omics, so please check it.

R: In this case, we refer to NifH protein sequences. So, we deleted the names of the primers for these genes to avoid confusion.

Page 11:

“the under sampling” -> “the undersampling”

 R: Changed

“in this study” -> “in this study,”

 R: Modified

“For that, over six late summer periods” -> “For that, over six late summer periods,”

 R: Modified

“These new rates recorded in Chile Bay are similar as those reported by (11) (44 nmol N L-1 d-1) with the exception that the higher rates recovered here were under dark conditions.” -> “These new rates recorded in Chile Bay are similar to those reported by (11) (44 nmol N L-1 d-1), with the exception that the higher rates recovered here were under dark conditions.” (here “as” should be “to” and “,” before “with the exception” would help the reader.

 R: Modified

Page 12:

“nitrogen, facilitated by the process of biological N2 fixation.” Please, remove “,” after nitrogen. Otherwise, the “facilitated” may modify the subject of the main clause.

 R: Modified

“we conducted a study in Chile Bay that, for the first time, reveals new knowledge about diazotrophy in the coastal seawaters of the WAP.” -> “we conducted a study in Chile Bay that, for the first time, revealed new knowledge about diazotrophy in the coastal seawaters of the WAP.”

  R: Modified

“In this bay” -> “In this bay,”

  R: Modified

Figures: The resolution is not as high as it could be. Maybe it is just the version I got. If there is room for increasing the resolution, I suggest increasing the resolution to maximize the impact of this great paper.

 R: The resolution of some figures was improved.

Reviewer 3 Report

See the file attached

Author Response

Review 3 of manuscript # 1526919

Light and dark diazotrophy during the summer in surface wa- ters of Chile Bay,

West Antarctic Peninsula. by María E. Alcamán-Arias et al.

The manuscript reports surface N2 fixation rates measured in situ in a coastal

environment in the West Antarctic Peninsula, during six consecutive austral summer periods, and discusses their contribution to primary production.

General comments

The work is original as it first reports summer N2 fixation rates in Antarctic surface coastal waters.

The manuscript is overall clearly written. The introduction is synthetic, yet it clearly presents the context, with appropriate support from the literature, as well at the objectives of the study.

The methodology in section 2.3 is too briefly exposed (see the detailed comments). For instance, references are indicated as for the way 15N and 13C incorporation rates were derived from the raw data, when a detailed procedure should be provided instead.

R: Methodology was improved adding much more details.

I also missed the main purpose of conducting 24h dark conditions when studying the system in summer: the relevance of such procedure and the complementary information that dark incubations provide (and why results may be quite different from the uncovered incubations) need to be clearly exposed, justified and discussed.

R: Incubations were made in darkness simulating the 30m depth, where the light shortage is almost total, according to our PAR measurements for 2018. These measurements indicated that about 1% of PAR light reached maxima up to 20m depth (Alcamán-Arias et al 2021). As for the 24 h used for our calculations, they were used to express daily rates, considering a 24h day. For clarity, we have supplemented this with more information in the methodology section.

This study is based on a one or two discrete measurements per year for a few years in a row in Chile Bay, a system that authors qualify as "a characteristic nutrient-rich Antarctic marine system with significant variability in Chl-a concentrations throughout the summer " (p.11). Then, how representative of summer conditions or populations are these very discrete samples? I therefore wonder whether a likewise highly fluctuating primary production follows the Chl-a variability throughout the summer, and if the composition and dynamics of bacterial populations also fluctuates within and between seasons. If so, the reported differences between years may be within the same range of magnitude as intra-seasonal differences and should be discussed in line with the current knowledge on the biogeochemistry of that area.

Conclusions

should thus be drawn with care. Authors mention that their N2 fixation measurements are "part of a marine biogeochemical monitoring time series". There may thus be some more material, data, or published papers they could refer to in their discussion. In section 4.3, one reads " In this bay we have previously characterized the identity and dynamics of phytoplankton, bacterioplankton and virioplankton under different productivity conditions (42,68,77), evaluated the microbial contribution to relevant ecological functions (42) ". I think the present results should be discussed in light of the knowledge drawn from these previous studies. In the present state, the discussion tends to draw quick causal conclusions from statistical correlations between very few data.

R: Unfortunately, it was not possible to study and therefore compare microbial communities between seasons in Chile Bay, and it was only possible to sample between summers. We have added more details of the Bay under study in the discussion section along with references describing the processes of changes in phytoplankton and bacterioplankton composition under contrasting chlorophyll conditions, where it is observed that the bacterioplankton community is less diverse under low chlorophyll conditions. In addition, the functional activity in the assimilation of nitrogenous compounds and inorganic C is decreased in the non-flowering periods, with the bacterial component being responsible for 100% of the N incorporations, and the diatoms for C fixation. In addition, other evidence of changes in the relative abundance and metabolic activity of the total microbial community under environmental changes such as low salinity due to local glacial melt was incorporated.

The goal of this study is to determine the contribution of N2 fixation to the new and local primary production. The positive rates of H13CO3- incorporation in samples incubated in the dark shown in Table 2 demonstrate that there is chemo-autotrophy in this system but this result is insufficiently brought forward from the beginning and the term "chemo-autotroph" does not even appear in the manuscript. The actual meaning of equation (a), in this case, could be discussed with an emphasis on the fact that this is a two-component system in which chemo-autotrophy represents a significant fraction of the total primary production. In the manuscript, I am thus missing some distinctions between primary production in the bacterial compartment (chemo-autotrophy) and that that coupled to photosynthesis (photo-autotrophy). The discussion should be rewritten with this concept as main story line.

R: The reviewer is correct. For clarification, we have included more detail and further discussion of chemoautotrophic activity in carbon fixation. Unfortunately, we do not have rates of carbon fixation by photoautotrophs the 24 h incubations, as uncovered bottles incubated at 2 m for 24 h contain both photosynthetic (approx.18 h sunlight) and chemoautotrophic (6 h dark) activity. However, we consider these light incubations as phototrophic, and the dark incubations as chemoautotrophic. Accordingly, we managed to estimate that in general chemoautotrophy could represent more than 30% of the total carbon fixation, representing even for the year 2017 more than 50%. Thanks to the reviewer's comments, we have included and discussed these results in a better way in the results and discussion section.

Another question insufficiently addressed here is whether and how N2 fixation may support primary production. Since all the identified diazotrophs are heterotrophs, N2 fixation directly supported bacterial growth, far more likely than the phytoplanktonic production. But N2 fixation would directly support chemo-autotrophic production if the two processes were coupled, i.e. if operating in the same organisms. Were the chemoautotrophs the identified diazotrophs? If not, N2 fixation is decoupled from inorganic carbon fixation in both the bacteria and the phytoplanktonic compartments... So any support from N2 fixation to the system could only be indirect, via exudation or recycling of the fixed N.

R: The referee is right. We supplemented this information with a bibliographic search of the references associated with the best heats of our sequences present in our phylogenetic tree (Fig 2). An explanation was added to the methodology and results-discussion section. From these results, only 5.6% of the reference genomes used here were found to correspond to chemoautotrophic organisms (Supplementary Table S1). Therefore, all previous information concerning the contribution of N2 fixation to PP was eliminated from the results and discussion in this review. The new results were added to clarify this part. Thus, we conclude that the N2 fixation process is supplementing bacterial growth but not phytoplankton growth, being partially uncoupled from the carbon fixation process according to the results of metabolisms associated with the genomes related to our recovered nifH sequences.

Detailed comments

The absence of line numbers doesn't facilitate references to specific words... I could only refer to page numbers and sections.

R: We apologize, but it seems that this problem must be associated with the presentation through the platform, since the original article had the line numbers. We hope that the platform will now be able to display this line.

Introduction

– p. 1. "Nitrogen is a limiting nutrient in most vast ocean regions". Consider replacing "vast" with "open".

R: Modified

– p. 1. "...biologically transforms atmospheric N2 into bioavailable ammonium (NH4+), which is important for assessing the global role... ". The verb "assessing" refers to "essential process": odd sentence, please rephrase.

R: Rephrased

– p.1 "...predictions estimate that N2 fixation rates could contribute up to 3.5 Tg N y-1 to the nitrogen budget ". Reference missing to support this statement

R: Reference was added.

– p. 2 "...thereby affording a better understanding". Consider replacing "affording"with "allowing"

R: Changed

Methods

– There was a change in probes and analytical protocols between years 2013-2016 and 2016-2019. Have cross-validation verifications been operated to make sure that neither the change in probes (e.g. salinity) or protocols (e.g. Chl-a) were responsible for possibly observed differences?

R:  The reviewer is right. Physical measurements of temperature and salinity were taken with multiparameter for some years and CTD for others, and therefore the data must be taken with caution when comparing between years. The chlorophyll measurements were also made in different years with 2 different, but widely used and validated methods, and are shown with the same unit of measurement. In this case, the measurements between 2017-2019 were made with the most sensitive method demonstrated in literature that was cited in the text.

– p 2-3 "For these late summers..." which ones? All of them? Is this part of the sentence really necessary?

R: Rephrased

– p. 4 " ...uncovered and covered to simulate light and dark conditions, respectively. The incubation time for both uncovered and covered bottles was for 24-25 hours ...". This second sentence suggests there were two sets of 3 bottles (one covered and theother not) incubated for about 24h, meaning that one triplicate was left covered for 24h and the other not? Yes. A better explanation about the experimental setup was added to the method section What kind of material was used to cover the bottles and did it prevent any light from reaching in? What was the purpose of this dual incubation procedure - Please explain first what hypothesis/question required the incubation of covered samples. Also, if samples were retrieved at a coastal site, why were they incubated off-shore and not at the site of sampling? Why were the 30m samples incubated at the surface (please explain)? Was the daily cycle of temperature (and not only that at the time of sampling) at 30m at the sampling site the same as that at the incubation point at 3m depth? Please rephrase this section and develop in further details.

R: We added more details to this method section that hopefully now clarify all doubts.

– p. 4 " adding 2 mL of 15N2 gas (98% atom 15N2 gas; Sigma-Aldrich)" Did you check for 15NO3 and 15NH4 contamination in this gas?

R: We greatly thank all the reviewers for their comments and criticisms regarding the possible contamination of the Sigma N2 gas vials, and apologize for the errors made in this manuscript.

Trying to understand the magnitude of the error in the measured rates, we corroborate that indeed for the measurements performed in this study between the years 2013 and 2014, Sigma-Aldrich brand N2 gas batch SZ1670 was used. Therefore, these rates must indeed be taken with great caution, due to possible contamination. However, we verified that the vials of N2 gas used from 2016 to 2019 were fortunately purchased from Cambridge Isotope Laboratories, Inc. whose manufacturing has shown no reported contamination to date. These have now been clarified, and new information was added to the manuscript. Thus, the rates recorded by our work since 2016 are under the expected standards in the use of 15N2 gas stocks. New information was added in the methods section and in discussion about this possible error due to contamination in the rates obtained during the first years of study (2013-2014) when using the 15N2 Sigma batch, reported by Dabundo et al., 2014 and Böttjer et al., 2017. 

These rates from 2016 were now the only ones actually discussed in this study, and were similar to those previously reported (44 nmol N L-1 d-1; (reference 11)) around ice-covered regions in the Southern Ocean, with the exception that the highest rates recovered here were under dark conditions. Our rates were also similar to those already reported for other illuminated open-water areas of the Southern Ocean (33,34) and Arctic Ocean (10,31), as now described in the manuscript.

Unfortunately, we cannot make the corrections that the referee requests because we no longer have the original N2 gas batches used for the years in question. For this reason we cannot, for example, test the gas dilution curves in water or evaluate possible contamination of the standards of reference used in the IRMS, among other corrections mentioned in the final work of White et al., 2020.

– p. 4 Filtering the incubated water through 0.7 μm GF/F filter may have led to partial loss of bacteria. Given that diazotrophs identified in this study were bacteria, the measured 15N incorporation may be underestimated.

R: Yes, the reviewer is right, but we used standard IRMS protocols that used 0.7 μm GF/F filters because smaller GFF filters do not exist.

On the other hand, to reveal the identity of diazotrophs we filtered the samples by Sterivex filters of 0.22 um. So, we get the whole picophytoplankton community for DNA material extractions.

– p.4, section 2.3 " The contribution of marine N2 fixation to NP was calculated according to (43), and included annual nitrate assimilation rates but excluded the nitrification rate correction." Please detail the methodology and calculation steps. Did the mass spectrometer provide absolute, atom% of13C and 15N enrichment, or d15N values?

R: The steps for N2P calculation are explained in the formula describe in the manuscript. The formula to 15N2, 13C, and 15NO3 calculation rates also are referenced to previous work cited in the ms. The IRMS provides both atom% of 13C/15N and delta.

– p. 4, section 2.3 " nitrogen (15N2) and carbon (13C) rates were calculated along with the mgC:mgN (C:N) ratio " Why was the C:N ratio in the biomass expressed in mg and not in molar ratio? Please specify whether this is the C:N ratio at the end of the incubation and whether a pre-incubation sample was taken to estimate the C:N ratio before the incubation.

R: Our mistake. The correct expression is molC:molN, now this was modified in the manuscript. We have molC:molC as natural abundance and at the end of the incubations as well.  This information was clarified now in the ms.

Please also specify which C and N contents were used to estimate the 15N and 13C incorporation. Please add this information after the sentence " Daily assimilation rates of 15N and 13C were calculated as described by (41,42), assuming a full day (24 h) of activity. "

R:  The paragraph has been modified for clarification.

– p.4, section 2.3 "The contribution of fixed nitrogen to primary production (PP)" Shouldn't this real "LPP" as in equation (a) and Table 2?

R: Yes, our mistake. Now this was changed along the manuscript for clarification.

– p.4, section 2.3 equations (a) and (b) (Shouldn't they be numbered instead?).

R: Yes, Thanks. The numbers were changed

Information is missing regarding these equations. Please explain what rN2fix, r13C and r15NO3 are, and develop how they were calculated.

R: The material and methods was modified for clarification.

Discussion

– p. 11 " In Chile Bay, the average N:P ratio recorded [...] represent a stable environment in terms of balanced stoichiometry" Consider revising your statement. The N:P ratio actually fluctuated between 9.45 and 18.22, which is a wide fluctuation around the Redfield value, and over a few time points only. Also, concluding on the stability of the nutrient stoichiometry based on one or two data points per year is farfetched...This first statement seems also contradictory with the end of the paragraph in which a shift from lower to higher N:P ratios (hence variable ratios...) in Antarctic coastal waters is discussed.

R: The referee is right. Now, we modified the sentence for clarification.

– p. 11 " the high rates in darkness may even be relevant during the dark winter months of this region." This statement requires further support from the literature. Are similar diazotrophic populations present and active in winter? I am missing the link between the observed occurrence of high rates in the dark in summer with the production that may be expected in winter. Said differently, efficiency in the dark in summer doesn't necessarily imply that N2 fixation may proceed in the dark winter... Consider reformulating.

R: The referee is right. There is no way with the available data that we can suggest that. Now we modified the sentence for clarification.

– p. 12 "Here, dark N2 fixation, such as that detected in the 2 m samples during the summers of 2016 and 2019, may contribute >37% to new nitrogen in the surface waters of Chile Bay." Covering the samples prevented photo-autotrophy to occur. In this regard, the reported contribution of diazotrophs to > 37% of new production is artificially increased, while the corresponding contribution in the unaffected samples is far lower. The relevance of this procedure and the suitability of the conclusion drawn from it should be argued and discussed.

R: We estimated the contribution only on dark fixation because our goal was to demonstrate that this process is relevant in these waters, as there was no sign of photoautotrophic organisms with diazotrophic capacity. We have now reorganized the manuscript and discussion for clarity.

– p. 12 " (where the highest N2 fixation rates were reported in 2018), we estimate that this process could fix ≥2.5 g N m-2 y-1" Here again, the highest rates were obtained in covered samples. . How representative is that of the natural system in summer? Extrapolations of N2 fixation in the whole bay should be performed using the rates obtained in the uncovered samples.

R: This calculation is an integration of the water column up to 30m in darkness, where more than 2 depths are needed to integrate.  We have N2 fixation rates at 2m, 25m and 30m in the dark, so it could only be done for these rates, since for incubations in light we only have 2m depth. Accordingly, it is not possible to estimate the contribution of N2 fixation in light periods.

– p. 12 "Inorganic nitrogen compounds are known inhibitors of diazotrophic activity (5). However, there is increasing evidence that biological N2 fixation may not be as sensitive to inorganic nitrogen concentrations" . An analysis of the effect of DIN on the metabolism of diazotrophic cyanobacteria recently came out in Microorganisms.

R: Many thanks. The reference was added and discussed.

– p. 12 "especially when phosphorous is not limited" Replace "limited" with "limiting" and "phosphorous" with "phosphorus" (for the latter, check all occurrences)

R: Corrected

– p. 12 "Consequently, the limiting effect of the phosphorus concentration on nitrogen fixation under light conditions could be associated with the growth of diazotrophic photoautotrophs, while..." I do not understand this conclusion as no autotrophic diazotroph was observed in this study. Please clarify and also conclude with caution as correlation does not necessarily imply causal relationship. This whole section is very speculative.

R: We rephrased the whole section for clarification.

– p. 12-13 "These high rates agree with the high carbon assimilation rates obtained especially under light conditions. [...] also in agreement with the chlorophyll concentrations." Please clarify... what do you mean by "agree with" here? If N2 fixation is operated by heterotrophic bacteria, it may not necessarily correlate with inorganic carbon incorporation, which is partly operated by the photo-autotrophic biomass.

R: The referee is right; we reformulate the paragraph.

– p. 13 "The fluctuation of nitrogen and carbon rates throughout these summers is also in agreement with the chlorophyll concentrations during each period." Add "fixation" between "carbon" and "rates" ? Please develop further this sentence. In which way is the N2 fixation activity (likely due to heterotrophic bacteria) in agreement with Chl-a (i.e. the concentration of photo-autotrophic biomass)? These sentences leave the probably wrong impression that N2 fixation directly supports the production of phytoplankton.

R: The referee is right; we reformulate the paragraph as mentioned above.

– p. 13 "N2 fixation may be necessary to maintain primary production in nutrient-rich coastal systems of Antarctica". I may have missed the point here... H13CO3- incorporation in the covered samples points to chemo-autotrophy. That is, part of the total primary production occurs within the bacterial compartment. but how are these chemo-autotrophs supported by N2 fixation? Have the diazotrophs identified in this study been described as being chemo-autotrophs? If not, chemo-autotrophy and N2 fixation are performed by different organisms within the bacterial community... it remains unclear how the latter would directly contribute to the production of the former.

R: A bibliographic search of the closest references to our sequences in the phylogenetic tree (Fig 2) was carried out, where only 5.6% of sequences with reference genomes corresponding to chemoautotrophic organisms could be found (Supplementary Table S1). Therefore, we have removed everything referring to the results and discussion of the contribution of N2 fixation to PP, and we have added new information explaining these new results to clarify this part.

We have also incorporated to the main text, that the N2 fixation process is supplementing bacterial growth and not that of phytoplankton, being partially uncoupled from the carbon fixation process according to genome sequences related to our own nifH sequences recovered.

– p. 13 "Interestedly, rates in Chile Bay were remarkably high in the dark, pointing to heterotrophic organisms as the most probable diazotrophs in the system ". Please be careful when using the expression "in the dark" as it usually refers to the dark phase of the light cycle. To avoid confusion, please be more explicit and state, e.g. "in samples maintained in permanent darkness" (even though the expression is lengthier) or "in covered samples".

R: The expression was changed to avoid confusions.

– p. 13 " Finally, analysis of the nifH gene enabled us [...] from glaciers near the bay as a result of runoff due to melt (65)." I would suggest to move this whole paragraph to the beginning of the discussion. This information is critical as it identifies who is most likely processing N2 fixation in this area and allows stating in which compartment (phototrophic or heterotrophic) it is occurring. This is needed to get a deeper understanding of the production data.

R: Now the discussion section was restructured according to all referee’s suggestions.

– p. 13 "contributing up to 37% of the new production in the nitrogen-based system, and representing a contribution of ≥2.5 g N m-2 y-1 ". See my remark above regarding this same statement.

R: Corrected

Figure 1. How comes the 0 depth level isn't aligned with the end of the blue color bar?

R: Corrected 

And is Fig3 oriented in the same way as the wider map in the insert? Please make sure they are.

Table 1 & 2. Since triplicate samples were taken, please specify the measured concentrations ± standard deviation for Chla and nutrients (Table 1) as well as for the rates of N2 fixation, 13C incorporation, and 15N2 contribution to total LPP and NP.

R: Now standard deviations were added to each variable that had replicates.

Round 2

Reviewer 1 Report

I appreciate that the authors have greatly revised their manuscript, mainly in response to the detailed critiques from the other reviewer. I also appreciate that they cannot go back in time and correct for the use of potentially contaminated gas, and thank them for being more transparent about their methodology in the revised manuscript. That said, more is needed - the authors provide no assessment of the limit of detection or minimum quantifiable rates for the N2 fixation rates, which cannot be omitted. This is really important when reporting N2 fix in new regions.

Another major concern is the nifH amplicon data. Several things need discussion/clarification: First, it is difficult to deduce but based on the tree figure, it seems like the number of reads recovered from this combined sample were actually extremely low for an Illumina study. Why is this? Were there lots of non-nifH sequences that were amplified? I see this commonly when there are not many nifH genes present in the environment, but this doesn't make sense given the very high rates reported for this year. This needs to be reported/discussed. Second, definitive statements about the metabolic lifestyle of the detected nifH-containing organisms based on being "closely related" to a cultivated diazotroph with a known metabolism, without reporting the percent identify to the reference sequence, is not appropriate. This makes parts of the discussion overly speculative.   

Repeated references to Shiozaki et al., 2020 are warranted, but the authors do not acknowledge that their findings have been contested in a recent "Matters Arising". 

Author Response

Referee 1

I appreciate that the authors have greatly revised their manuscript, mainly in response to the detailed critiques from the other reviewer. I also appreciate that they cannot go back in time and correct for the use of potentially contaminated gas, and thank them for being more transparent about their methodology in the revised manuscript. That said, more is needed - the authors provide no assessment of the limit of detection or minimum quantifiable rates for the N2 fixation rates, which cannot be omitted. This is really important when reporting N2 fix in new regions.

R: We thank the reviewer very much for their comment and understanding. We have now added to the materials and methods section the detection limit ranges in mg N and mg C (0.005 mg N and 0.074 mg C) derived from the IRMS used to calculate the cell incorporation quantifications. According to the values of the calibration curve of the equipment, our rates are above the detection limit.

Another major concern is the nifH amplicon data. Several things need discussion/clarification: First, it is difficult to deduce but based on the tree figure, it seems like the number of reads recovered from this combined sample were actually extremely low for an Illumina study. Why is this? Were there lots of non-nifH sequences that were amplified? I see this commonly when there are not many nifH genes present in the environment, but this doesn't make sense given the very high rates reported for this year. This needs to be reported/discussed.

R: From the raw sequencing data, we proceeded to trim the reads for quality and then the amino acid sequences were curated against HMM models described in NifMAP to remove nifH homologs (bchL, chlL, bchX, parA). For the PolF-PolR primer set, this process left a total of 2077 final reads that could be assigned (out of initial 4948 reads, 58% of these reads were removed after filtration). In the case of the Ueda19F-R6 primer set, 1377 reads were obtained that could be assigned from the initial 2830 reads (64% of these reads were removed after filtration). This information was added to the results section for clarification.

 Second, definitive statements about the metabolic lifestyle of the detected nifH-containing organisms based on being "closely related" to a cultivated diazotroph with a known metabolism, without reporting the percent identify to the reference sequence, is not appropriate. This makes parts of the discussion overly speculative.   

R: A BLASTP search was made to obtain the percentage of identical matches and e-value between the chemoautotrophs and the Chile Bay OTUs of NifH proteins. The % coverage between sequences obtained from this study and the references were added to Supplementary Table S1. 

Repeated references to Shiozaki et al., 2020 are warranted, but the authors do not acknowledge that their findings have been contested in a recent "Matters Arising". 

R: We appreciate the reviewer's input regarding this article. We added these references in the discussion section.

Reviewer 2 Report

I see a comprehensive revision by the reviewers, which did improve the presentation significantly. Thus, I have only minor/optional comments. I wish to express my great respect for the authors conducting years-long observations in low-temperature regions, resulting in an impressive outcome.

L 602-605:

In my previous comment, “ At the same time, I agree with the potential advantage of photosynthesis as it provides C. A modeling study shows that the ratio of resource C:N is important in the occurrence of N2 fixation. The author may cite the following paper to support their argument:

Inomura K, Bragg J, Follows MJ (2017) A quantitative analysis of the direct and indirect costs of nitrogen fixation: a model based on Azotobacter vinelandii. ISME Journal 11:166–175.”

I should have suggested as it shows the relationship between organic carbon input (relative to ammonium) and the rate of nitrogen fixation: 

Inomura K, Bragg J, Riemann L, Follows MJ. A quantitative model of nitrogen fixation in the presence of ammonium. PLoS ONE 2018; 13: e0208282. 

That said, both references are relevant in this context. 

L 741: After this, authors may discuss the result a bit, relating to O2. e.g., “It is possible that lower N2 fixation rate during the daytime may be due to higher concentrations of O2 based on photosynthetic activities.” However, I understand the authors' point “In this manuscript we decided not to discuss in depth the negative effect of O2 on N2 fixation, because we did not recruit any nifH sequence from phototrophic diazotrophs. Instead” So, I will leave the authors to decide.

L 1057: I wonder if [Internet] is needed?

Fig. 4 (new manuscript)

Here I copy the previous comment and the authors’ response:

“Regarding Figure 5, would it be possible to add panels for NO3-/PO43- for the x axis? I think people are interested in the rate of N2 fixation in relation to N:P of the nutrient. This information seems there in table one so I think I could be done simply?”

“ R: We don’t have Figure 5, maybe the referee is referring to Figure 3? We modified all the figures and Tables.”

I am sorry that I did not write down the correct figure number. Yes, I meant Figure 3. Also, I was trying to say PO43-/NO3- instead of NO3-/PO43-.  I see that in the new figure, I still see PO4-3 for x axis. Have you thought about plotting the y axis values against PO43-/NO3-? This is not a requirement, but there might be an interesting trend that authors may consider other figure panels or for a supplementary figure.

Author Response

Referee 2

 see a comprehensive revision by the reviewers, which did improve the presentation significantly. Thus, I have only minor/optional comments. I wish to express my great respect for the authors conducting years-long observations in low-temperature regions, resulting in an impressive outcome.

 R: Thank you very much for your positive and constructive comments.

L 602-605:

In my previous comment, “ At the same time, I agree with the potential advantage of photosynthesis as it provides C. A modeling study shows that the ratio of resource C:N is important in the occurrence of N2 fixation. The author may cite the following paper to support their argument:

Inomura K, Bragg J, Follows MJ (2017) A quantitative analysis of the direct and indirect costs of nitrogen fixation: a model based on Azotobacter vinelandii. ISME Journal 11:166–175.”

I should have suggested as it shows the relationship between organic carbon input (relative to ammonium) and the rate of nitrogen fixation: 

Inomura K, Bragg J, Riemann L, Follows MJ. A quantitative model of nitrogen fixation in the presence of ammonium. PLoS ONE 2018; 13: e0208282. 

 That said, both references are relevant in this context. 

R: We have added the new references (96 and 97 in the reference list) as suggested by the reviewer.

L 741: After this, authors may discuss the result a bit, relating to O2. e.g., “It is possible that lower N2 fixation rate during the daytime may be due to higher concentrations of O2 based on photosynthetic activities.” However, I understand the authors' point “In this manuscript we decided not to discuss in depth the negative effect of O2 on N2 fixation, because we did not recruit any nifH sequence from phototrophic diazotrophs. Instead” So, I will leave the authors to decide.

R: We very much appreciate the reviewer's comments, and we understand and agree with the importance of this possible effect of oxygen on the diazotrophic community in these waters. However, we have decided not to go deeper into this possible effect because, as explained above, we did not obtain any sign of photosynthetic diazotrophs in these samples and we unfortunately do not have dissolved oxygen concentration measurements for all the years sampled. On the other hand, we still do not know the life style of these diazotrophic heterotrophs (i.e., whether they live in association with particles, pellets, suspended sediments, etc). Therefore, we believe that further data collection during future research is needed to discuss this effect with more support and detail. 

L 1057: I wonder if [Internet] is needed?

R: Our apologies, it is an error of the reference editor. We have revised everything to eliminate this type of error. 

Fig. 4 (new manuscript)

Here I copy the previous comment and the authors’ response:

“Regarding Figure 5, would it be possible to add panels for NO3-/PO43- for the x axis? I think people are interested in the rate of N2 fixation in relation to N:P of the nutrient. This information seems there in table one so I think I could be done simply?”

“ R: We don’t have Figure 5, maybe the referee is referring to Figure 3? We modified all the figures and Tables.”

I am sorry that I did not write down the correct figure number. Yes, I meant Figure 3. Also, I was trying to say PO43-/NO3- instead of NO3-/PO43-.  I see that in the new figure, I still see PO4-3 for x axis. Have you thought about plotting the y axis values against PO43-/NO3-? This is not a requirement, but there might be an interesting trend that authors may consider other figure panels or for a supplementary figure.

R: Thank you for the suggestion. We have made a new figure relating the fixation rates to the N:P ratio shown in Table 1. However, as when we plotted the rates in relation to nitrogen, the obtained result was not significant with R2<0.3. For this reason, we have decided to show the reviewer this figure, but not to include it in the manuscript, and therefore to keep Figure 3 as it is. We hope that the reviewer will agree with this decision.

Reviewer 3 Report

Authors have provided a thoroughly revised version of their manuscript and provided rather detailed answers to my questions;  I find the overall story much clearer. The discussion was deeply reorganized and is taking new routes as it now more explicitly develops on the different metabolic strategies of diazotrophs in regard to the observed nifH-based diversity.

The presentation of the methodology is also much clearer. In particular, I now understand why samples had to be covered, to simulate the actual irradiance experienced at 30m depth. There remain however two aspects of this methodological strategy that are insufficiently explained in the text:

i/ What was the difference in temperature between the offshore, 2m deep sampling point, 30m deep sampling point, and the 2m deep, onshore incubation point? If there are differences between these three points, how significant are they?  Is this ought to affect the C and N incorporation activities and the resulting, estimated production rates? Could a bias be introduced because of this temperature difference?

ii/ How well mixed is the upper 30m (i.e. how deep is the surface mixed layer)? If homogeneous, one may expect that populations at 2 or 30 m are actually stirred within the water column and, over a 24h time window, they may overall experience the same average or cumulated irradiance. Therefore, the clear distinction between the ambient conditions observed at 2m and 30m need to be supported by local physical measurement (or appropriate references) indicating that the upper column on this very spot is strongly stratified. Looking at the Table on p 12, I do not see any strong vertical gradient in either T, NO3, or PO4 for instance that would suggest the surface layer is stratified.  I am missing, in the introduction, a more explicit description of this stratification and of the physical dynamics of the water masses in the Bay, which would motivate and justify the relevance of discriminating between the depts of 2 and 30m. This may be even more important that Table 1 indicates that measurements at 2 and 30m depth were not taken on the same days... I am wondering whether the observed differences (in the values of a given parameter) between 2m and 30m may actually reflect temporal fluctuations more than spatial differences.

If a clear vertical stratification cannot be asserted, then authors may want to re-consider the analysis and discussion of their results, possibly by pooling depths and focusing on the comparison between phototrophic vs chemotrophic populations of diazotrophs instead of comparisons per depth?

Author Response

Referee 3

Authors have provided a thoroughly revised version of their manuscript and provided rather detailed answers to my questions; I find the overall story much clearer. The discussion was deeply reorganized and is taking new routes as it now more explicitly develops on the different metabolic strategies of diazotrophs in regard to the observed nifH-based diversity.

 The presentation of the methodology is also much clearer. In particular, I now understand why samples had to be covered, to simulate the actual irradiance experienced at 30m depth. There remain however two aspects of this methodological strategy that are insufficiently explained in the text:

i/ What was the difference in temperature between the offshore, 2m deep sampling point, 30m deep sampling point, and the 2m deep, onshore incubation point?

R: On average the temperatures found at 2 m and 30 m were 0.8°C and 1.2°C, respectively. At the surface point where the experiments were performed, the temperature was between 1-1.5°C.This information is now included in the text for clarification (Lines 188 and 253)

If there are differences between these three points, how significant are they? 

R: The differences are not significant. Despite the shallowness of the incubation site, the air and wind temperatures remain constant and cool in ranges of 1-1.5°C.

 Is this ought to affect the C and N incorporation activities and the resulting, estimated production rates? Could a bias be introduced because of this temperature difference?

R: As mentioned above, there are no significant differences between the incubation and in situ temperatures at each depth. Therefore, this is not expected to influence the estimated N and C fixation rates.

ii/ How well mixed is the upper 30m (i.e. how deep is the surface mixed layer)? If homogeneous, one may expect that populations at 2 or 30 m are actually stirred within the water column and, over a 24h time window, they may overall experience the same average or cumulated irradiance. Therefore, the clear distinction between the ambient conditions observed at 2m and 30m need to be supported by local physical measurement (or appropriate references) indicating that the upper column on this very spot is strongly stratified. Looking at the Table on p 12, I do not see any strong vertical gradient in either T, NO3, or PO4 for instance that would suggest the surface layer is stratified.  I am missing, in the introduction, a more explicit description of this stratification and of the physical dynamics of the water masses in the Bay, which would motivate and justify the relevance of discriminating between the depts of 2 and 30m. This may be even more important that Table 1 indicates that measurements at 2 and 30m depth were not taken on the same days... I am wondering whether the observed differences (in the values of a given parameter) between 2m and 30m may actually reflect temporal fluctuations more than spatial differences.

If a clear vertical stratification cannot be asserted, then authors may want to re-consider the analysis and discussion of their results, possibly by pooling depths and focusing on the comparison between phototrophic vs chemotrophic populations of diazotrophs instead of comparisons per depth?

R: In a recently published article (Alcaman-Arias et al., 2022), we discussed the stability of the water column for the years 2017-2019 in Chile Bay. In our previous study, we concluded that there are periods of surface stratification down to 10 m depth, as well as periods of mixing as a consequence of strong seasonal winds. However, no deep water intrusion could be found in the bay by T-S plot analysis. According to the referee's comments we have rephrased the results section regarding the physical and biological variables in Table 1, emphasizing that these values correspond to the times when the seawater was collected at 2 and 30 m. Likewise, we have included in the discussion the above-mentioned reference with the intention of reporting the water column states for at least the years 2017-2019.

On the other hand, we only detected heterotrophic and chemoautotrophic organisms distributed among the first meters of the water column (30 m), but not photosynthetic diazotrophic organisms; thus, it is not possible to make a more in depth analysis between both groups of organisms in these waters for the times sampled here.
